# Dynamic changes in brain lateralization correlate with human cognitive performance

Xinran Wu[1][ᵒ], Xiangzhen Kong[2][ᵒ], Deniz Vatansever[1][ᵒ], Zhaowen Liu[3], Kai Zhang[4], Barbara J. Sahakian[1,5,6], Trevor W. Robbins[1,5,7], Jianfeng Feng[1,8,9,10], Paul Thompson[11], Jie Zhang [1] *

1 Institute of Science and Technology for Brain-Inspired Intelligence, Fudan University, Shanghai, China, 2 Department of Psychology and Behavioral Sciences, Zhejiang University, Zhejiang, China, 3 Psychiatric & Neurodevelopmental Genetics Unit, Center for Genomic Medicine, Massachusetts General Hospital, Boston, Massachusetts, United States of America, 4 School of Computer Science and Technology, East China Normal University, Shanghai, China, 5 Department of the Behavioural and Clinical Neuroscience Institute, University of Cambridge, Cambridge, United Kingdom, 6 Department of Psychology, University of Cambridge, Cambridge, United Kingdom, 7 Department of Psychiatry, University of Cambridge School of Clinical Medicine, Cambridge, United Kingdom, 8 Department of Computer Science, University of Warwick, Coventry, United Kingdom, 9 Collaborative Innovation Center for Brain Science, Fudan University, Shanghai, China, 10 Shanghai Center for Mathematical Sciences, Shanghai, China, 11 Imaging Genetics Center, Mark & Mary Stevens Institute for Neuroimaging & Informatics, Keck School of Medicine, University of Southern California, Los Angeles, California, United States of America

ᵒ These authors contributed equally to this work.
* zhangjie80@fudan.edu.cn

**Data Availability Statement:** Raw fMRI data of Human Connectome Project and behavioral measurements are publicly available and can be downloaded from the Human Connectome Project

## Abstract

Hemispheric lateralization constitutes a core architectural principle of human brain organization underlying cognition, often argued to represent a stable, trait-like feature. However, emerging evidence underlines the inherently dynamic nature of brain networks, in which time-resolved alterations in functional lateralization remain uncharted. Integrating dynamic network approaches with the concept of hemispheric laterality, we map the spatiotemporal architecture of whole-brain lateralization in a large sample of high-quality resting-state fMRI data ($N = 991$, Human Connectome Project). We reveal distinct laterality dynamics across lower-order sensorimotor systems and higher-order associative networks. Specifically, we expose 2 aspects of the laterality dynamics: laterality fluctuations (LF), defined as the standard deviation of laterality time series, and laterality reversal (LR), referring to the number of zero crossings in laterality time series. These 2 measures are associated with moderate and extreme changes in laterality over time, respectively. While LF depict positive association with language function and cognitive flexibility, LR shows a negative association with the same cognitive abilities. These opposing interactions indicate a dynamic balance between intra and interhemispheric communication, i.e., segregation and integration of information across hemispheres. Furthermore, in their time-resolved laterality index, the default mode and language networks correlate negatively with visual/sensorimotor and attention networks, which are linked to better cognitive abilities. Finally, the laterality dynamics are associated with functional connectivity changes of higher-order brain networks and correlate with regional metabolism and structural connectivity. Our results provide insights into the adaptive nature of the lateralized brain and new perspectives for future studies of human cognition, genetics, and brain disorders.

Access (http://www.humanconnectome.org/data), while subjects' personal information (or "restricted data", like information about zygosity and parents, et al.) is restricted for researchers who meet the criteria for access to confidential data (https://www.humanconnectome.org/study/hcp-young-adult/document/1200-subjects-data-release). Researchers want to get the restricted data can contact HCP Project Manager according to the application process provided by HCP (https://www.humanconnectome.org/study/hcp-young-adult/document/restricted-data-usage) to obtain the data. Subject identification numbers in the database and all other relevant data are within its Supporting Information files (S1–S5 Data).

**Funding:** Data used in this work were provided by Human Connectome Project (https://www.humanconnectome.org/). JZ was supported by Science and Technology Innovation 2030 - Brain Science and Brain-Inspired Intelligence Project (Grant No. 2021ZD0200204), Shanghai Municipal Science and Technology Major Project (No.2018SHZDZX01) and NSFC 61973086, and ZJLab. XK is supported by the Fundamental Research Funds for the Central Universities (2021XZZX006), the National Natural Science Foundation of China (32171031), and Information Technology Center of Zhejiang University. DV was funded by the National Natural Science Foundation of China (No. 31950410541), the Shanghai Municipal Science and Technology Major Project (No. 2018SHZDZX01). PMT was supported, in part, by NIH grant U54 EB020403. JF was supported by the 111 Project (No. B18015), the key project of Shanghai Science and Technology (No. 16JC1420402), National Key R&D Program of China (No. 2018YFC1312900), National Natural Science Foundation of China (NSFC 91630314). KZ was supported by the Shanghai Pujiang Program. The funders had no role in study design, data collection and analysis, decision to publish, or preparation of the manuscript.

**Competing interests:** The authors declared that there are no competing interests.

**Abbreviations:** AFG, Arenas-Fernandez-Gomez; AI, autonomy index; ALFF, amplitude of low-frequency fluctuation; BCT, Brain Connectivity Toolbox; BMI, body mass index; CAB-NP, Cole-Anticevic Brain-wide Network Partition; DFC, dynamic functional connectivity; DLI, dynamic laterality index; DTI, diffusion tensor imaging; DZ, dizygotic; FA, fractional anisotropy; FD, frame distance; FDR, false discovery rate; FN, fiber number; FSL, FMRIB Software Library; GS, global signal; HCP, Human Connectome Project; LF, laterality fluctuations; LR, laterality reversal; MLI, mean laterality index; MZ,

## Introduction

Hemispheric lateralization is a prominent feature of human brain organization [1], with interhemispheric differences repeatedly observed in both structure and function [2–8]. For example, the left planum temporale, commonly referred to as the Wernicke's area, shows reliable activity in cognitive paradigms that probe auditory processing and receptive language [9,10]. In addition to such reports from "task-induced activation" studies on functional lateralization [7,11], emerging evidence also indicates lateralization within the human brain's intrinsic connectivity architecture at rest. For example, recent neuroimaging studies suggest that hemispheric lateralization, estimated using network-based approaches on resting-state fMRI data, can accurately predict activity-based lateralization during cognitive task performance [12,13]. Together, existing evidence alludes to the vital contribution of hemispheric lateralization to healthy and adaptive mentation.

Functional lateralization is traditionally considered as a static and trait-level characteristic of individuals [8,12,14,15] that is hypothesized to enhance neural capacity [16,17]. For example, laterality of language and attention networks have been previously associated with individual differences in linguistic and visuospatial abilities [8,12,14,15]. Beyond such trait-level characteristics, however, recent evidence also suggests the "dynamic" nature of intrinsic brain networks over time [18–20]. Dynamic functional connectivity (DFC) can track the time-varying alterations in cognitive states, task demands, and performance [21–23], providing insights into how brain network reconfiguration relates to cognition, consciousness, and psychiatric disorders [19,22,24,25]. In parallel, emerging findings now indicate that the degree of lateralization is instantaneously modulated by various external factors such as attention, task contexts, and cognitive demands [26–29], which may arise from time-varying interactions between bottom-up and top-down neural processing [7,30]. Therefore, it is possible that hemispheric lateralization also changes across time to accommodate changing demands of the environment, which may be evaluated by dynamic brain network approaches.

However, these 2 vital aspects of brain network interactions, in other words, the dynamic changes of brain lateralization and their relationship to higher-order cognition, have not been explored to date. Therefore, in the present study, we developed a measure of "dynamic lateralization" and tested its significance in explaining individual variability in cognitive performance. Specifically, we investigated the laterality dynamics by 2 complementary measures, i.e., laterality fluctuations (LF) and laterality reversal (LR), which reflect moderate and extreme changes in laterality, respectively, on intrinsic brain networks constructed from high-quality resting-state fMRI data from the Human Connectome Project (HCP) [31]. Here, we show that LF are positively associated with language function and cognitive flexibility, while LR shows an opposite effect, suggesting a balance between intra and interhemispheric information communication. Furthermore, negative correlations in time-varying laterality between default mode network and visual/sensorimotor and attention networks and their relationship with cognitive performance were revealed, suggesting parallel information processing capacity, which may facilitate adaptive cognition. Additionally, we also investigated the neural and anatomical factors that may affect dynamic laterality of the human brains and established the heritability of the dynamic laterality measures.

## Results

### Dynamic laterality index

We analyzed resting-state fMRI data from 991 participants in the HCP cohort and extracted BOLD time series of all 360 cortical regions using a group-level parcellation scheme (HCP

monozygotic; PALM, Permutation Analysis of
Linear Models; ROI, region of interest; SC,
structural connectivity.

MMP1.0) [32]. To map the time-varying lateralization architecture, we developed a measure termed dynamic laterality index (DLI) by calculating the laterality index in each sliding window for each region of interest (ROI) (Fig 1A). Specifically, we adopted a global signal (GS)-

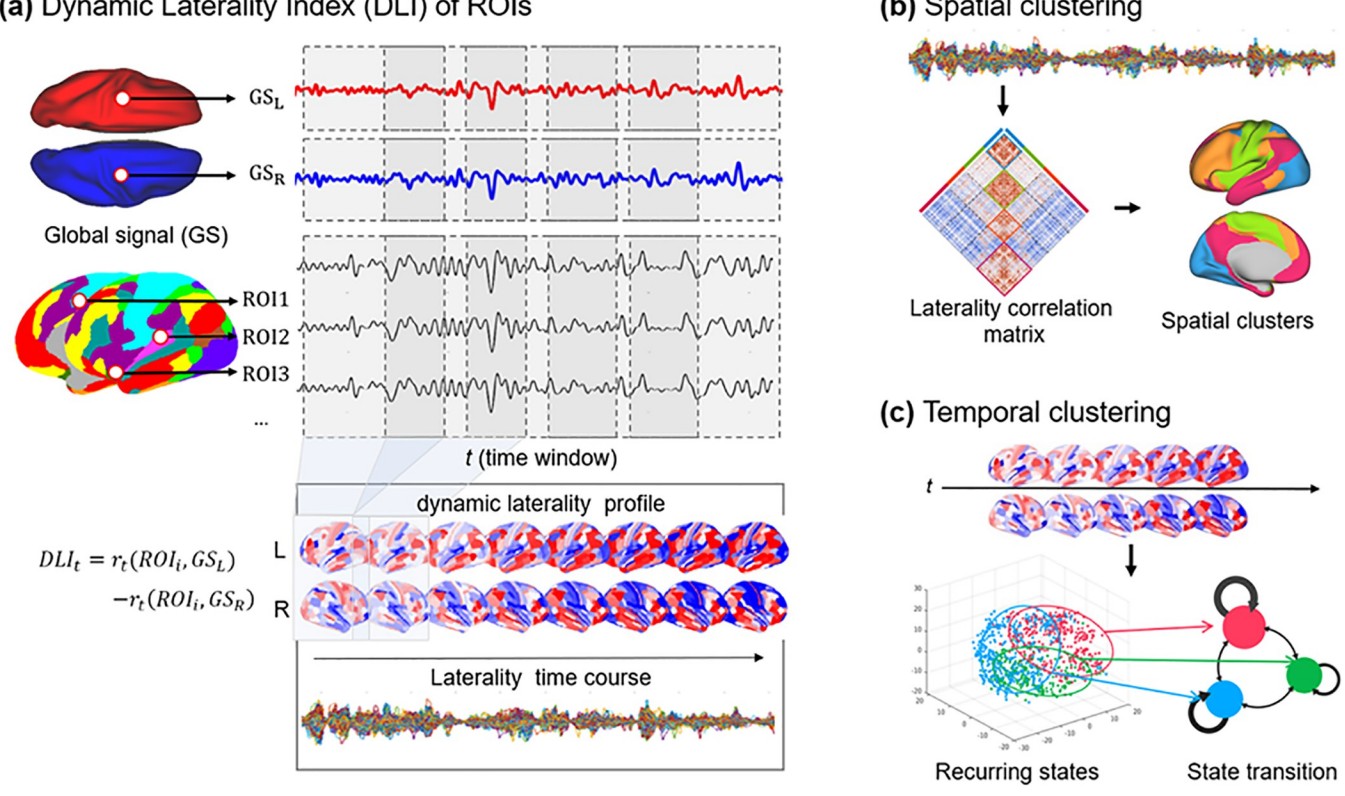

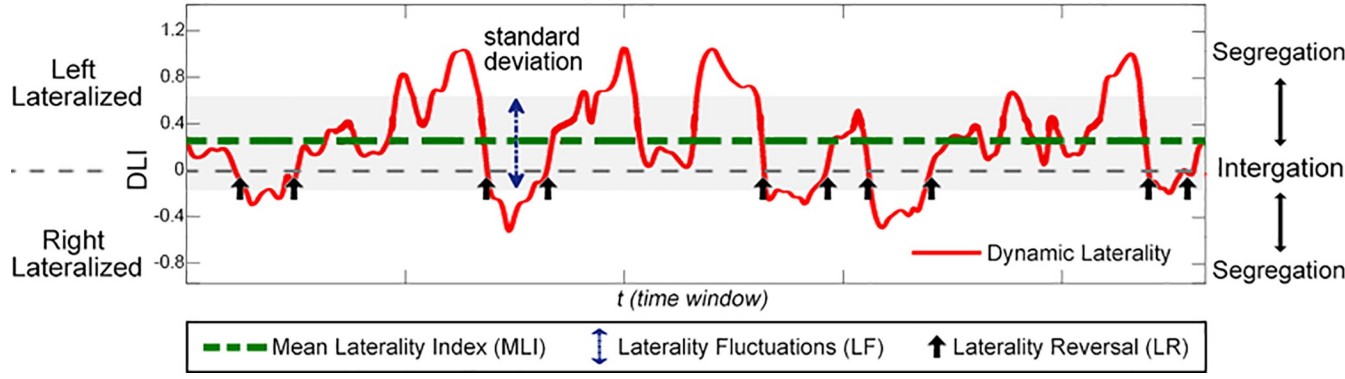

**Fig 1. The workflow of dynamic laterality analysis.** (a) Definition of the DLI. The DLI of $ROI_i$ within time window $t$ is defined as the correlation coefficient (z-transformed) between $GS_L$ and the ROI minus the correlation coefficient between $GS_R$ and the ROI. Using a sliding window approach, we obtained a time series of DLI for each ROI. (b) Laterality correlation matrix, which is obtained by correlating laterality time series across all ROIs. Spatial clustering is then performed to identify spatial clusters of brain regions demonstrating covariation in laterality time series. (c) Temporal clustering of whole-brain laterality patterns, which identifies potential recurring laterality patterns. (d) Illustration of DLI and relevant dynamic laterality measures using an ROI that is left-lateralized. The red curve represents the time series of DLI. The green dotted line is the MLI (0.26), and the blue double arrow denotes the standard deviation of the laterality time series, which measures the level of LFs. The black arrow represents the LR (the change of the sign of lateralization across 2 consecutive time windows). Large magnitude of laterality index indicates segregation at the hemispheric level, while small magnitude of laterality index (near 0) indicates integration across 2 hemispheres. DLI, dynamic laterality index; GS, global signal; $GS_L$, global signal of the left hemisphere; $GS_R$, global signal of the right hemisphere; LF, laterality fluctuation; LR, laterality reversal; MLI, mean laterality index; ROI, region of interest.

based laterality index that can effectively capture brain lateralization characteristics underlying higher-order cognition [14], defined as the difference between an ROI's BOLD correlation with the GS of left brain and its correlation with the right brain at each time window (see Methods). GS-based laterality index of an ROI reflects whether the activity of the ROI is more synchronized with the left or the right hemisphere. The correlation between BOLD signal of an ROI and the left/right hemispheric global signal ($GS_L$/$GS_R$) represents its synchronization with the left/right hemisphere, respectively. Higher correlation of an ROI with $GS_L$ compared to $GS_R$ indicates left-hemispheric lateralization and vice versa. DLI of an ROI (a time series of laterality index) is then obtained by calculating the laterality index for each time window; see Fig 1A. Positive DLI of a region indicates stronger interaction with the left hemisphere (i.e., leftward laterality), while a negative one indicates rightward laterality.

We demonstrate that the GS-based laterality index we adopted is highly similar to a conventional laterality measure autonomy index (AI) [12,33] that is widely used. Using resting-state fMRI data from HCP, we found high correlation coefficient between the whole-brain laterality profile obtained by the mean laterality index (MLI, average across all time windows) and that obtained by AI (Pearson r = 0.64 ± 0.12, all participants have $p < 0.05$). The MLI has good replicability over 4 sessions involving left and right scans (correlation across sessions: 0.66 ± 0.018); see S1 Fig. Furthermore, GS-based laterality index is efficient in that it only takes $2^*n$ time, compared to the traditional ROI-based method that takes $n^2$ time (n being the number of regions), therefore is economic for multiple time windows.

Using this DLI, we characterized the time-averaged laterality of large-scale brain networks in resting state. The left and right hemispheres largely illustrated positive and negative mean laterality, respectively (Fig 2A). In the left hemisphere, regions in the language (MLI = 0.17 ± 0.08) and default mode (MLI = 0.14 ± 0.07) networks showed strong leftward laterality (networks defined by Cole-Anticevic Brain-wide Network Partition (CAB-NP) [34]). In the right hemisphere, the cingulo-opercular (MLI = −0.11 ± 0.07), dorsal attention (MLI = −0.10 ± 0.06), and visual networks (MLI = −0.08 ± 0.08; Fig 2A) showed strong rightward laterality. Frontoparietal network illustrated strong laterality within both hemispheres (left: MLI = 0.14 ± 0.07, right: MLI = −0.15 ± 0.07). Comparatively, bilateral sensorimotor regions (left: MLI = 0.036 ± 0.04, right: MLI = −0.022 ± 0.05) and left visual areas (MLI = −0.025 ± 0.06) depicted relatively weak mean laterality.

We further characterized the dynamic changes in laterality of a region from 2 different perspectives: the magnitude and the sign of laterality, by LF and LR, respectively; see Fig 1D. LF is defined as the standard deviation of laterality time series. Standard deviation of a time-varying measure is commonly used in dynamic brain network analysis (e.g., DFC), which reflects its variability [24,35–37]. LR specifically refers to the number of zero crossings of laterality (switch between left and right laterality) in 2 consecutive windows. It is also inspired by the dynamic brain network analysis, i.e., a region may change the module it belongs to (named "nodal flexibility" proposed by Bassett and colleagues), which correlated closely with cognitive ability [38,39].

We illustrate laterality time series and these 2 measures using a brain region with high level of mean laterality (0.26, left laterality; Fig 1D). The standard deviation of time-varying laterality index corresponds to moderate changes, or deviations, with respect to the mean laterality. By contrast, LR reflects larger changes in laterality, corresponding to sufficiently extreme deviations from the mean, which results in sign changes in laterality. Moreover, a large magnitude of the laterality index corresponds to segregation at the level of hemispheres, and a small magnitude (near 0) indicates integration across 2 hemispheres; see Fig 1D and "Network and structural basis of laterality dynamics" (Results) for explanation.

Based on the above 2 dynamic laterality characteristics, we found that the laterality of most brain regions varied considerably over time. Specifically, primary visual (LF = 0.46 ± 0.09;

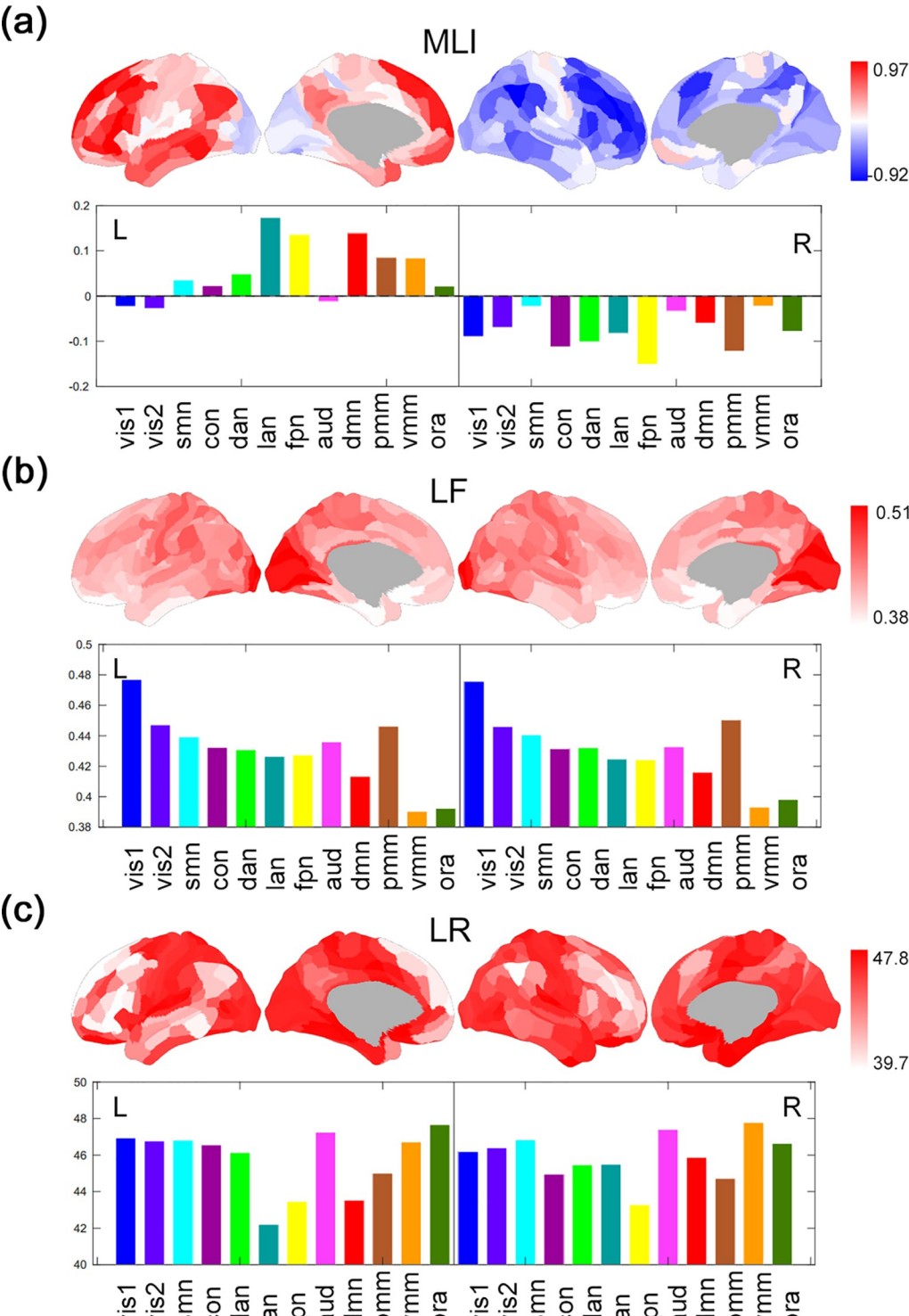

**Fig 2. Dynamic architecture of whole-brain functional lateralization.** (a) MLI, i.e., time average of DLI. (b) LFs, i.e., standard deviation of laterality time series. (c) LR, i.e., the number of switches in the sign of laterality (from positive to negative or vice versa) of all 360 brain regions (averaged across 991 participants), which are rearranged into 12 subnetworks by the CAB-NP. Vis1, Visual1; vis2, Visual2; smn, Somatomotor; con, Cingulo-Opercular network; dan, Dorsal-Attention network; lan, Language network; fpn, Frontoparietal network; aud, Auditory network; dmn, Default Mode network; pmm, Posterior-Multimodal; vmm, Ventral-Multimodal; ora, Orbito-Affective. L, left hemisphere; R, right hemisphere. The underlying data for Fig 2A can be found in S1 Data. The underlying data for Fig 2B can be found

in S2 Data. The underlying data for Fig 2C can be found in S3 Data. CAB-NP, Cole-Anticevic Brain-wide Network Partition; DLI, dynamic laterality index; LF, laterality fluctuation; LR, laterality reversal; MLI, mean laterality index.

LR = 46.8 ± 1.7) and sensorimotor regions (LF = 0.44 ± 0.09; LR = 46.6 ± 1.9) generally illustrated stronger levels of variation in laterality (both LF and LR) than higher-order association regions—including the frontoparietal (LF = 0.43 ± 0.11; LR = 43.4 ± 1.97), language (LF = 0.43 ± 0.10; LR = 43.5 ± 2.1), and default mode networks (LF = 0.41 ± 0.10; LR = 44.7 ± 1.7); see Fig 2B and 2C.

## Spatial clustering of the laterality dynamics

After characterizing the temporal LF across the whole brain, we investigated how spatially distributed regions that support different functional specializations correlate with each other in laterality (Fig 1B). Through spatial clustering of the laterality time series across all brain regions, we identified 4 major clusters (Fig 3A): Cluster 1 consisted mainly of regions from the bilateral visual network; Cluster 2 consisted of regions from the bilateral sensorimotor and cingulo-opercular networks; Cluster 3 mainly covered the frontoparietal network (more from the right hemisphere) and attention network; and Cluster 4 mainly consisted of regions from bilateral default mode network, part of the frontoparietal network (mainly the left hemisphere) and regions from the language network (Fig 3A). Clusters 1 and 2 showed higher levels of LF (reflected by higher standard deviation of DLI and LR) when compared to Clusters 3 and 4 (repeated-measures ANOVA, $p < 0.001$, pairwise $t$ test with $p < 0.01$; see S3 Fig).

We then explored the association between laterality dynamics and out-of-scanner cognitive performance. Specifically, we used multiple tasks adopted by HCP that are generally associated with either lateralized (e.g., language: story comprehension task, and attention: Flanker task) or bilateral (e.g., cognitive flexibility: Card Sort task, and working memory: List Sorting task) brain function. First, we found that the LF and LR of the identified spatial clusters correlated in a positive and negative manner, respectively, with cognitive performance. Second, significant correlation was only found for language function (story comprehension tasks), cognitive flexibility (Card Sort Test and pattern comparison task), and processing speed (Pattern Comparison Test) out of all 13 cognitive measures; see S1 Table. Correction for multiple comparisons were performed using the false discovery rate (FDR) method for all 29 laterality measures employed in our analyses (see Methods section for details of these laterality measures). Furthermore, we also adopted a more conservative multiple-comparisons correction scheme for the 13 behavioral measures. That is, based on the FDR correction of the 29 dynamic laterality measures (FDR q < 0.05), we further employed Bonferroni correction for the 13 behavioral measures, i.e., FDR q < 0.05/13 = 0.0038. All the following results were based on the FDR correction for all 29 laterality measures (FDR q < 0.05). Those correlations that passed the stricter correction method (FDR q < 0.05/13) were also marked.

Specifically, LF of Clusters 3 (FPN) and 4 (DMN and language network) correlated positively with the difficulty of stories a participant could understand (language task difficulty, LanDiff, Cluster 4: $r = 0.11$, $p = 0.018$; Cluster 3: $t = 0.11$, $p = 0.014$; Fig 3C). A cognitive flexibility measure (Dimensional Change Card Sort Test, CardSort) also correlated positively with LF across all clusters (Cluster 1, $r = 0.12$, $p = 0.001$; Cluster 2, $r = 0.14$, $p < 0.001$; Cluster 3, $r = 0.14$, $p < 0.001$; Cluster 4, $r = 0.14$, $p < 0.001$; all FDR q < 0.05/13). In contrast, LR of these clusters correlated negatively with cognitive performance: for language task difficulty, $r = -0.2$ ($p < 0.001$, FDR q < 0.05/13) for Clusters 3, and $r = -0.14$ ($p = 0.002$) for Clusters 4. For Card Sort task, $r = -0.15$ ($p < 0.001$, FDR q < 0.05/13) for Cluster 3 and $r = -0.15$ ($p < 0.001$, FDR q < 0.05/13) for Cluster 4. In addition, the performance of Pattern Comparison Processing

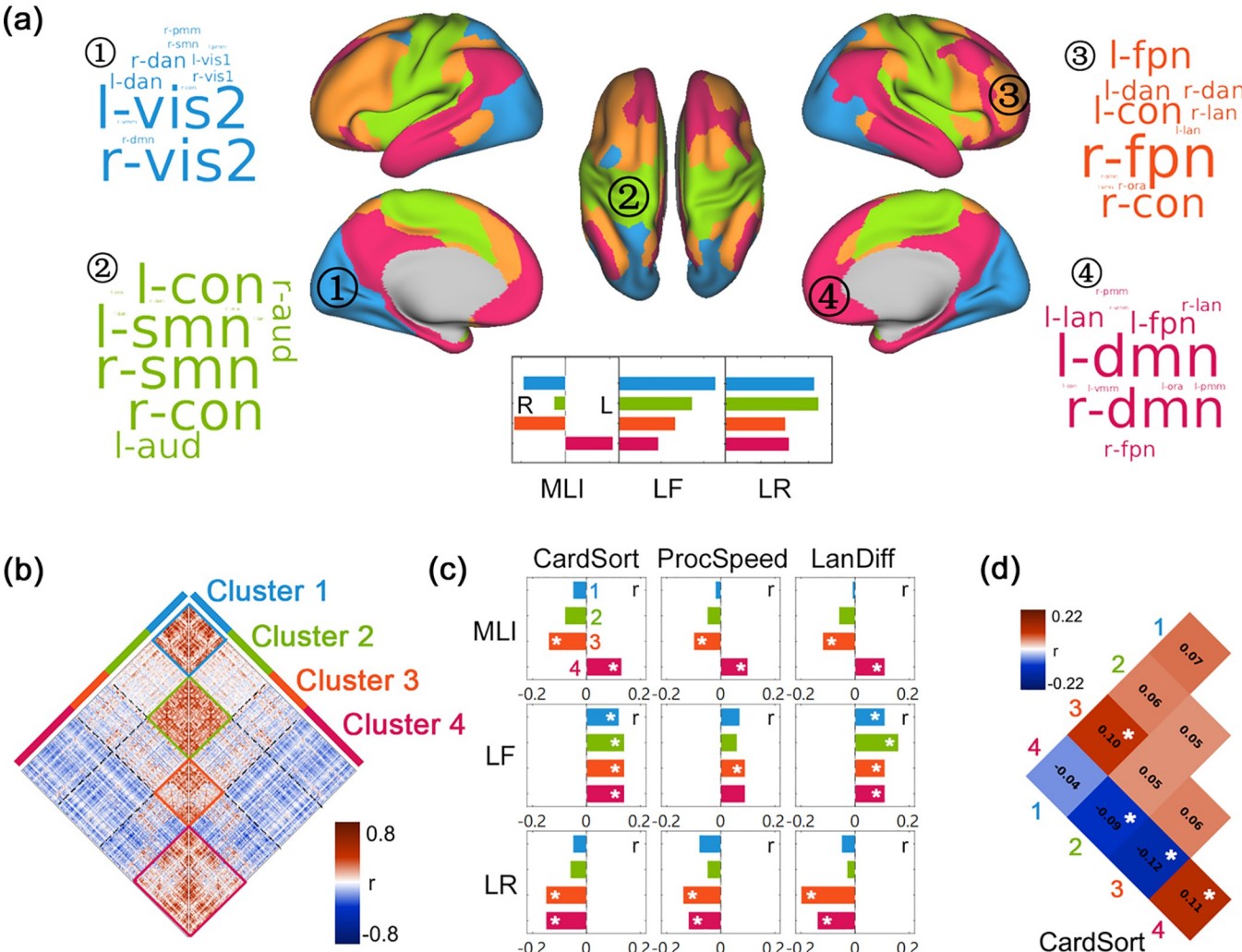

**Fig 3. Spatial clustering of laterality dynamics.** (a) Four spatial clusters revealed by clustering of the laterality time series of all 360 brain regions. The word cloud for each cluster indicates the functional networks being involved (based on the CAB-NP), with the font size representing the proportion of each network in the cluster. L, left-lateralized; R, right-lateralized; MLI, mean laterality index; LF, laterality fluctuations; LR, laterality reversal. The mean value of the 3 dynamic laterality measures of the 4 spatial clusters are shown in the inset. (b) The laterality correlation matrix across 360 brain regions (averaged over all 991 participants). (c) The correlation between the 3 dynamic laterality measures (MLI, LF, LR) of the 4 spatial clusters and cognitive performance of 3 tasks. CardSort, the performance of Dimensional Change Card Sort Test; ProcSpeed, the performance of Pattern Comparison Processing Speed Test; LanDiff, mean difficulty of stories for each participant in HCP language task. (d) The association between the laterality correlation within/between clusters and the processing speed (Pattern Comparison Processing Speed Test). Only significant results (Pearson correlation, FDR corrected, FDR q < 0.05, marked by one asterisk *) are shown. The underlying data for this figure can be found in S4 Data. CAB-NP, Cole-Anticevic Brain-wide Network Partition; FDR, false discovery rate; HCP, Human Connectome Project.

Speed Test (ProcSpeed, the speed of completing the task) also showed positive association with LF of Cluster 3 ($r = 0.09$, $p = 0.009$) and negative association with LR of Clusters 3 ($r = −0.14$, $p = 0.001$) and 4 ($r = −0.12$, $p = 0.003$).

We furthermore explored how time-varying laterality time series of different clusters correlate with each other. Importantly, Cluster 4 showed negative laterality correlations with all other 3 clusters (Cluster 1: $t = −81.7$; Cluster 2: $t = −84.37$; Cluster 3: $t = −62.1$, all $p < 0.001$; Fig 3B), while Clusters 1, 2, and 3 showed positive correlation among themselves (Cluster 1 and 2, $t = 20.4$; $p < 0.001$; Clusters 2 and 3, $t = 15.2$; $p < 0.001$), indicating that Cluster 4 shows a tendency to lateralize in the hemisphere opposite to those of the other 3 clusters over time.

The negative laterality correlation between Cluster 4 and Clusters 1, 2, and 3 also bear functional significance. We found that individuals with higher Card Sort score showed stronger negative correlation in laterality between Cluster 4 and Cluster 2 ($r = −0.09$, $p = 0.012$), and Cluster 4 between Cluster 3 ($r = −0.12$, $p = 0.001$, FDR q < 0.05/13); see Fig 3D and S2 Table.

## Temporal clustering of laterality dynamics

We further explored the temporal organization of laterality dynamics by clustering whole-brain laterality state of multiple time windows (see Fig 1C and Methods). Three recurring laterality states (Fig 4A) were identified, each showing distinct laterality patterns, different dwelling times, and transition probabilities (Fig 4B). Specifically, State 1 showed a typical

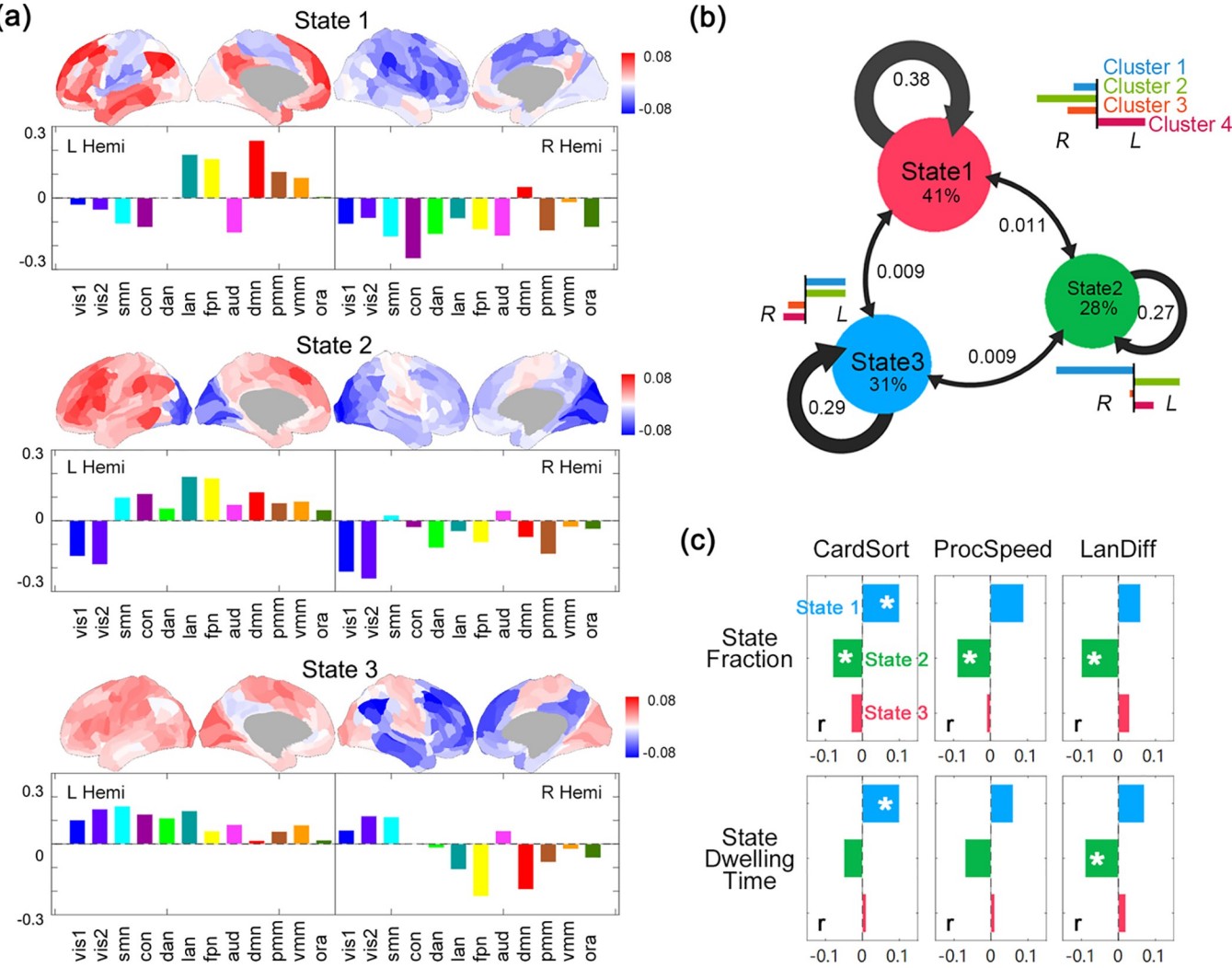

**Fig 4. Temporal clustering of laterality dynamics.** (a) The 3 recurring laterality states obtained by temporal clustering of the time-varying, whole-brain laterality states (360 regions) with colors indicating the network arrangement of the CAB-NP. L, left hemisphere; R, right hemisphere. (b) The mean fraction of 3 states and the mean probability of switching between them. The bar plot next to each state represents the averaged laterality of the 4 spatial clusters in each state. L, left-lateralized; R, right-lateralized. (c) The correlation between the fraction/mean dwelling time of 3 states and memory ability (mem)/language ability. (d) The correlation between the probability of switching between/within the 3 states and language ability. Only significant results (Pearson correlation, FDR corrected, FDR q < 0.05, marked by one asterisk *) are shown. The underlying data for this figure can be found in S4 Data. CAB-NP, Cole-Anticevic Brain-wide Network Partition; FDR, false discovery rate.

leftward laterality in Cluster 4 ($t = 77.9$, $p < 1e$-20) and a rightward laterality in Cluster 2 ($t = -109.3$, $p < 1e$-20) (Fig 4A and 4B). State 1 is the primary laterality state with the largest fraction of 41% and a mean dwelling time of 49.9 ± 3.58 windows (about 35 s). State 2 showed a typical rightward laterality in Cluster 1 ($t = -108$, $p < 1e$-20), and a leftward laterality in Cluster 2 ($t = 60.1$, $p < 1e$-20) and Cluster 4 ($t = 53.2$, $p < 1e$-20) (Fig 4A and 4B). State 3 showed a leftward laterality in Cluster 1 and 2, and rightward laterality of Cluster 4 ($t = -64.7$, $p < 1e$-20) and Cluster 3 ($t = -41.4$, $p < 1e$-20). These 2 states showed similar fractions (State 2: 28%; State 3: 31%) and dwelling time (State 2: 10.5 ± 2 s; State 3: 12.1 ± 2.3 s).

We further investigated the behavioral correlates of individual variations in the state transition properties. Results showed a link between State 2 and cognitive functions: Individuals with less State 2 showed higher language task difficulty (fraction, $r = -0.1$, $p = 0.007$; dwelling time, $r = -0.09$, $p = 0.015$), higher CardSort performance (fraction, $r = -0.08$, $p = 0.014$) and higher ProcSpeed score (fraction, $r = -0.09$, $p = 0.01$). In addition, CardSort was positively correlated with the fraction ($r = 0.1$, $p = 0.006$) and dwelling time ($r = 0.1$, $p = 0.014$) of State 1. See Fig 4C and 4D and S3 Table. Multiple comparisons correction was performed using the FDR method for all laterality measures used in the analyses; see Methods section.

## Network and structural basis of laterality dynamics

Next, we investigated how dynamic laterality is related to functional and structural brain network properties. First, we investigated the functional correlates of dynamic laterality, i.e., we investigated how dynamic laterality of a cluster is related to the time-varying functional connectivity of the brain. We then calculated the correlation between time-varying laterality index and time-varying network features (including degree, participation coefficient, and modularity, reflecting brain integration/segregation) and BOLD activity (amplitude of low-frequency fluctuation (ALFF)).

We found that the functional connectivity of the high-order brain networks, especially those in the left hemisphere (e.g., FPN, DMN, and language network, mainly covered by Cluster 4) play an essential role in dynamic changes in lateralization. For the 3 right-lateralized clusters (Clusters 1/2/3), higher level of right lateralization was associated mainly with weaker functional connectivity of these clusters with the higher-order networks on the left hemisphere (Fig 5B). For Cluster 4 that is intrinsically left-lateralized, its increased left lateralization was accompanied by reduced functional connectivity with multiple lower-order networks at the right hemisphere, including many of the networks covered by Clusters 1/2/3 (the right subfigure of Fig 5B). Stronger functional connectivity of a cluster with its ipsilateral higher-order networks also plays a role in the increased lateralization of the 4 clusters but has less contribution (Fig 5B; details can be found in S4 Fig).

For functional network features and ALFF, we found that the whole-brain lateralization (the mean absolute value of DLI across all 360 regions) correlated negatively with the averaged degree ($r = -0.43 ± 0.1$, 91% $p < 0.05$), participation coefficient ($r = -0.2 ± 0.12$, 81% $p < 0.05$), interhemispheric connection ($r = -0.52 ± 0.11$, 92% $p < 0.05$), and ALFF ($r = -0.22 ± 0.12$, 85% $p < 0.05$), while showing positive correlations with modularity of the brain network ($r = 0.47 ± 0.06$, 91% $p < 0.05$) across time windows; see S5 Fig. This indicates that when the whole brain is highly segregated (modular), it also demonstrates high laterality, while an integrated state of the whole brain (low modularity) are accompanied by low laterality. At the local level, we found that these correlation patterns are most prominent for the high-order brain regions (most significantly in default mode network, the language network, and the frontoparietal network; S5 Fig).

Second, we explored the structural correlates of laterality dynamics using diffusion MRI data. We used diffusion imaging data from a subset of HCP S1200 (HCP Unrelated 100) to

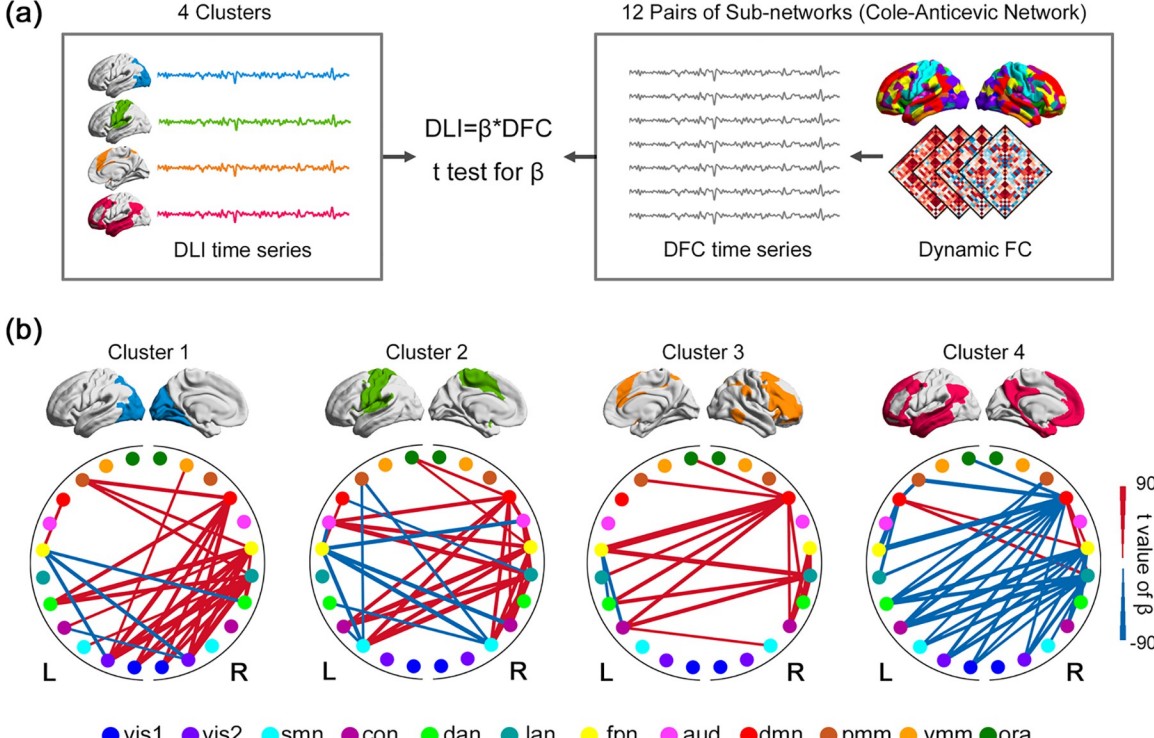

**Fig 5. Association between dynamic laterality time series of the 4 identified clusters and the DFC within/between 24 subnetworks.**
(a) Analysis pipeline. We established linear regression models of cluster-level DLI time series on subnetwork-level DFC and performed one-sample *t* test on the regression coefficient (β) to investigate whether the influence of DFC on DLI was significantly greater than 0 (positive coupling) or less than 0 (negative coupling). (b) The association between dynamic laterality time series of Clusters 1/2/3/4 and the DFC within/between 24 subnetworks (according to CAB-NP). Only functional connectivity with very significant *t*-value of regression coefficient (with $p < 1e\text{-}200$ or $|t| > 40$, one-sample *t* test) is being shown since there are too many significant connections. Here, a positive laterality index indicates left lateralization. Colors of points indicate the network arrangement of the CAB-NP. L, left hemisphere; R, right hemisphere. The underlying data for this figure can be found in S4 Data. CAB-NP, Cole-Anticevic Brain-wide Network Partition; DFC, dynamic functional connectivity; DLI, dynamic laterality index; aud, Auditory network; con, Cingulo-Opercular network; dan, Dorsal-Attention network; dmn, Default Mode network; fpn, Frontoparietal network; lan, Language network; ora, Orbito-Affective; pmm, Posterior-Multimodal; smn, Somatomotor; Vis1, Visual1; vis2, Visual2; vmm, Ventral-Multimodal.

constructed 2 kinds of structural connectivity matrices, fractional anisotropy (FA) matrix and fiber number (FN) matrix for each participant. We found significant positive correlations between the laterality correlation matrix and structural connectivity matrix, including FA matrix (Spearman's *rho* = 0.14 ± 0.02, significant in 99% participants) and FN matrix (Spearman's *rho* = 0.14 ± 0.02, significant in 99% participants); see Fig 6A. In line with this, homotopic regions generally showed positive correlations in their laterality time series (mean r = 0.39 to 0.79; S2 Fig) due to the critical role of corpus callosum [40]. These results suggested that brain regions with similar laterality dynamics tend to have stronger structural connection. Furthermore, LF of a region correlated positively with its FA/FN-degree, i.e., the sum of FA/FN of fibers between this region and all other regions (LF-FN: *rho* = 0.32 ± 0.09, significant in 99% participants; LF-FA: *rho* = 0.28 ± 0.08, significant in 99% participants; see Fig 6B).

## Heritability of laterality dynamics

Finally, we investigated heritability of lateralization dynamics by twin analyses (using the kinship information). First, we estimated heritability through the ACE model [additive heritability

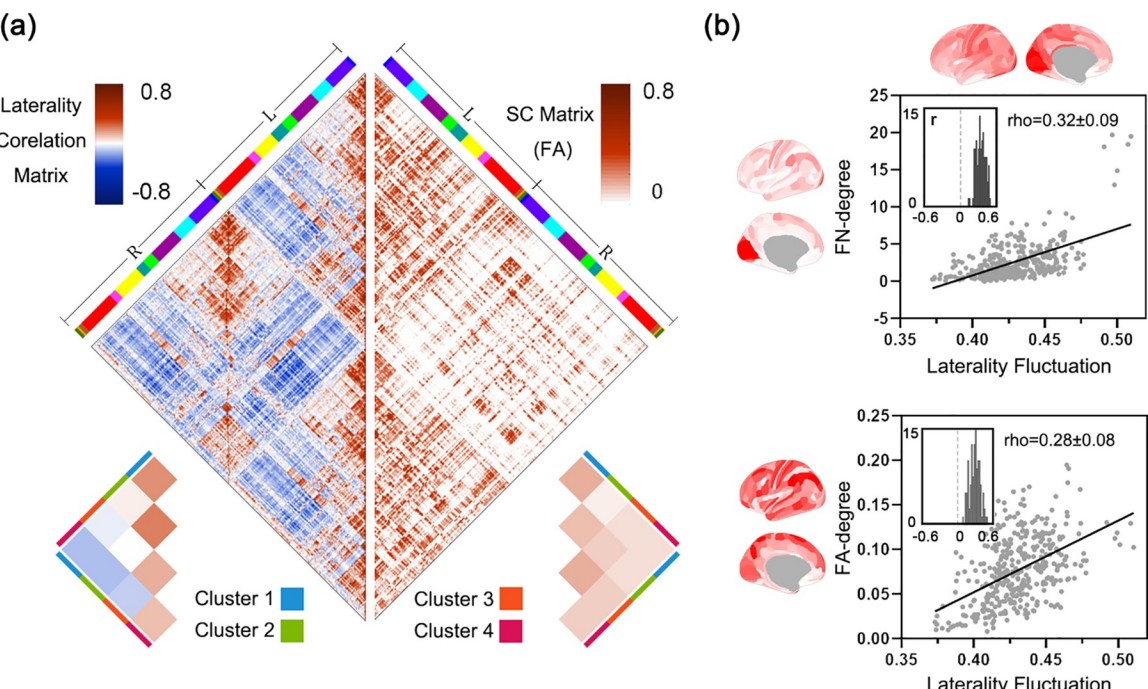

**Fig 6. Structural basis of dynamic laterality.** (a) Laterality correlation matrix (left half-matrix) and structural connection matrix (right half-matrix) demonstrates high level of correlation. These 2 matrices are obtained by averaging across all 99 participants. L, left hemisphere; R, right hemisphere. The color ribbons indicate the network arrangement of CAB-NP detailed in Fig 2. (b) Upper: the relationship between a structural measure (FN-degree) and the LF across all brain regions. Lower: the relationship between another structural measure (FA-degree) and the LF. The large scatter maps show the association between the averaged map of dynamic laterality measures and the averaged FA-/FN-degree map (both across all 99 participants), and the inset shows the distribution of the individual correlation coefficients between the FA-/FN-degree map and laterality fluctuation map of each participant. The underlying data for this figure can be found in S4 Data. CAB-NP, Cole-Anticevic Brain-wide Network Partition; FA, fractional anisotropy; FN, fiber number; LF, laterality fluctuations.

(A), common (C), and specific (E) environmental factors model]. We found that LF showed the highest heritability ($h^2$) among all dynamic laterality measures ($h^2 = 0.09$ to $0.48$ for LF; $h^2 = 0.002$ to $0.41$ for MLI; $h^2 = 0.001$ to $0.31$ for LR; see Fig 7A–7C). Of all 4 spatial clusters, Cluster 4 (e.g., the default mode network, part of the right frontoparietal network, and the language network) showed the highest heritability (MLI $h^2 = 0.19$; LF $h^2 = 0.4$; LR $h^2 = 0.06$), followed by Cluster 3 (MLI $h^2 = 0.19$; LF $h^2 = 0.36$; LR $h^2 = 0.06$). The heritability of Cluster 2 (MLI $h^2 = 0.13$; LF $h^2 = 0.35$; LR $h^2 = 0.03$) and Cluster 1 (MLI $h^2 = 0.12$; LF $h^2 = 0.33$; LR $h^2 = 0.03$) were relatively low.

We then calculated the cosine distance between each pair of monozygotic (MZ) twins, dizygotic (DZ) twins, sibling (SI), and unrelated group participants using various dynamic laterality measures across the whole brain. As expected, the laterality correlation matrix showed greater similarity within MZ twins than those within DZ or non-twins (one-way ANOVA, $F = 152.7$, $p < 0.0001$; Fig 7D). Similarly, the similarity of MLI and LF (of all regions) in twins, SIs, and unrelated participants showed a decreasing trend, although the difference between MZ and DZ was not significant (for MLI, $F = 23.8$, $p < 0.0001$, MZ versus DZ, $p > 0.99$; for LF, $F = 10.8$, $p < 0.0001$, MZ versus DZ, $p = 0.99$; Fig 7A and 7B). There was no significant difference in LR between the 4 groups ($F = 1.48$, $p = 0.22$; Fig 7C). All one-way ANOVA were done using GraphPad Prism 9.3.1 (http://www.graphpad.com). In summary, dynamic laterality measures were generally heritable.

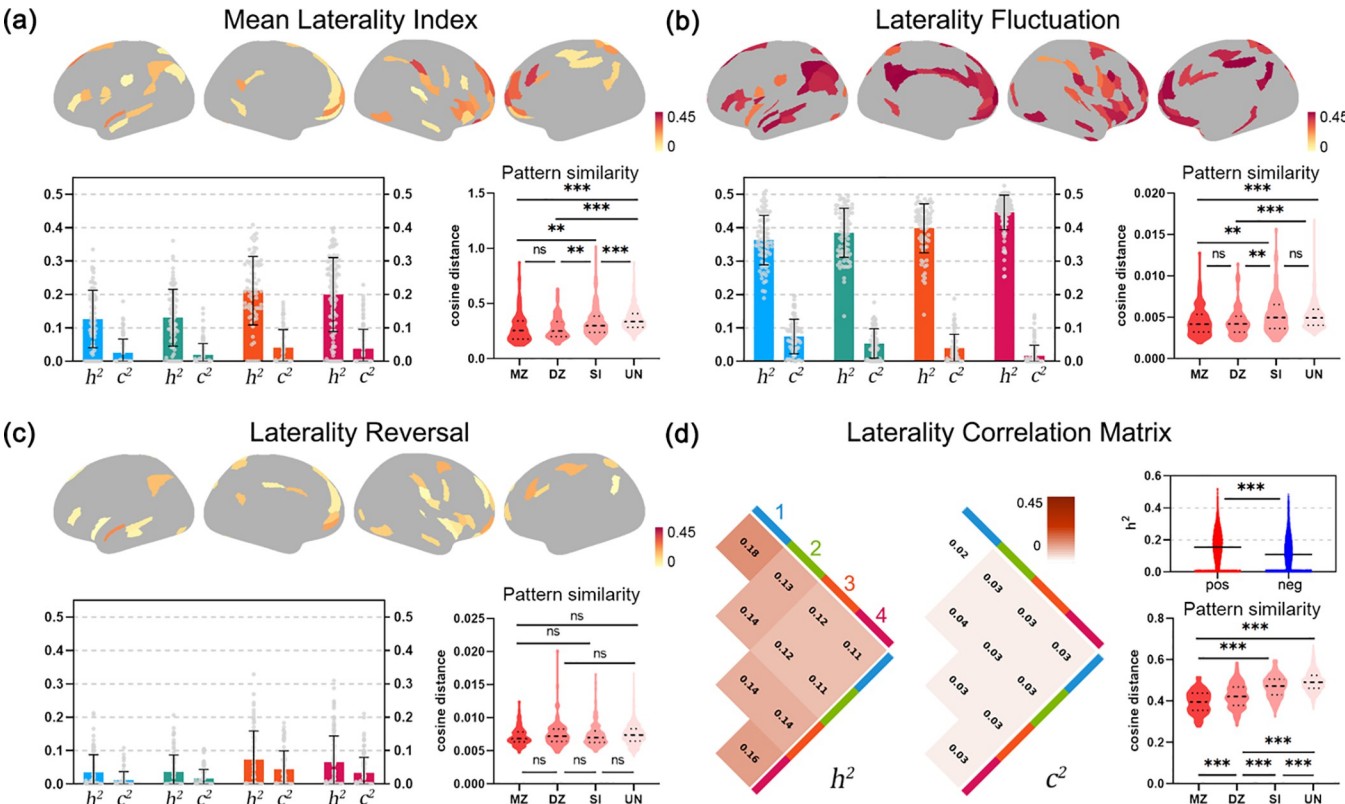

**Fig 7. Heritability of dynamic laterality.** (a-c) Heritability analysis of MLI, LF, and LR. Upper: Heritability ($h^2$) of each brain region. Only the regions with $h^2$ of $p < 0.05$ are retained. Lower left: The $h^2$ and common environmental factors ($c^2$) for each cluster, and each data point represents a brain region. Lower right: The cosine distance between whole brain map for each pair of MZ twins, DZ twins, SI, and UN. Asterisk indicates the significant difference between the groups (one-way ANOVA). (d) Heritability of lateralization correlation matrix. Left: Average $h^2$ and $c^2$ in 4 clusters. Upper right: Differences in heritability between the positive and the negative links, i.e., links with positive or negative correlation in laterality. Lower right: The cosine distance of laterality correlation matrix of each pair of MZ, DZ, SI, and UN. The error bar represents the mean value ±1 SD. One asterisk (*), $p < 0.05$; 2 asterisks (**), $p < 0.01$; 3 asterisks (***), $p < 0.001$; ns, nonsignificant. Pos, positive correlation; neg, negative correlation. The underlying data for this figure can be found in S4 Data. DZ, dizygotic; LF, laterality fluctuations; LR, laterality reversal; MLI, mean laterality index; MZ, monozygotic; SI, sibling; UN, unrelated group.

## Validation analysis

To rule out the possible influence of potential nuisance variables on our dynamic laterality results, we performed a series of validation analyses: (1) the effect of sample size: all 991 participants were split into two halves, and the main analyses were repeated separately on both halves of the data (S10 Fig); (2) head movements: we repeated the main analyses with multiple head movement parameters (Friston 24 parameters [41]) being regressed out from the BOLD signal (S12 Fig). We also used more stringent exclusion criteria (mean frame distance (FD) < 0.15 mm and FD < 0.1 mm) to repeat our analyses (S12 Fig); (3) GS: we repeated the main analyses, while regressing out the GS of the whole brain (S11 Fig). This resulted in the sum of the GS from the left and right hemispheres being 0, i.e., when the $GS_L$ is positive, the $GS_R$ is equal to its negative number (rather than a GS of 0 for one hemisphere). Therefore, the correlation coefficients could still be calculated between hemispheric GS and individual ROIs; (4) time window lengths: another 2 window lengths (60 TRs and 90 TRs) were used to investigate the reproducibility of DLI (S13 and S14 Figs); and (5) the potential contaminating effect of an ROI to its ipsilateral GS: when we calculated the correlation coefficient between an ROI and the GS of its ipsilateral hemisphere, the ROI was included in this calculation, which may have "contaminated" its ipsilateral GS. We therefore removed the ROI when extracting the ipsilateral

hemisphere GS and repeated our analyses to examine whether our results are affected (S15 Fig). In all the above cases, our main results, including the distribution of dynamic laterality indices across the whole brain, and their correlation with cognitive performance, were largely replicated, which indicated the reliability of the DLIs. See more details in S1 Text.

## Discussion

Laterality is traditionally argued to represent a stable, trait-like feature of the human brain. However, considerable amount of work has recently been devoted to characterizing DFC that has been shown to be related to mental flexibility [19] and to predict ongoing mental states or task performance [21], which inspires the question of whether laterality of the brain is also fluctuating over time. This study proposed a framework for dynamic laterality analysis. The time-averaged DLI replicated well previous findings in terms of the static spatial patterns of functional lateralization [12–14], while the time-varying laterality measures provided new insights into the dynamic changes of lateralization in resting state. Specifically, different levels of temporal variation of laterality were found across the brain: Regions within the visual and sensorimotor networks generally showed higher variations, while regions from the default mode, frontoparietal, and language network showed relatively weaker variations. This may suggest that laterality in lower-order brain regions is more flexible, consistent with previous research showing that lower-order regions have shorter intrinsic time scale than higher-order regions [42]. That is, the bilateral sensory areas need to process constantly changing sensory inputs and integrate them with information from the contralateral hemisphere in real time to form accurate and coherent percepts.

To systematically characterize dynamic laterality changes, we used 2 measures, i.e., LF and LR, which showed positive and negative correlations with cognitive performance, respectively. These opposing associations suggest that these 2 measures capture different aspects of dynamic laterality. Generally, brain regions in the left and right hemisphere illustrated positive and negative mean laterality, respectively (indicating that they have more ipsilateral connections), with their laterality fluctuating around the mean value (Fig 1). Previous literature has suggested that comprehending literal meanings was associated with left language areas (high left laterality), while dealing with difficult metaphors were shown to involve the right homotopic regions [43]. Greater fluctuations of laterality therefore suggested larger number of states of intra and interhemisphere interactions, which may involve both left hemisphere language areas and the right homotopic regions that may be beneficial for difficult language tasks. For cognitive flexibility (Card Sort task) that generally involves both hemispheres [44] and larger fluctuations of laterality suggested flexible recruitment of both hemispheres that is potentially beneficial.

Although larger fluctuations in laterality correlated with better cognitive performance, extreme changes in laterality, i.e., frequent LR correlated negatively with task performance. LR suggests that laterality of a region fluctuate too much and deviates remarkably from the mean value (the dominant laterality regime; Fig 1D), i.e., a region's functional connectivity switches from being more ipsilateral to highly contralateral. Frequent reversal thus indicates that a brain region constantly leaves its dominant functional regime (i.e., more ipsilateral connectivity). Collectively, these results suggest that moderate changes in laterality may enhance cognitive performance, while extreme laterality changes may hamper cognition. In the Results part, we have shown that low laterality of the whole brain corresponds to high level of interhemispheric communication (or less intrahemispheric information processing), while high laterality of the whole brain corresponds to the opposite case; these results therefore indicate optimal cognitive performance may be related to a dynamic balance between interhemispheric information exchange and intrahemispheric information processing. As we also show that low

laterality of the whole brain corresponds to low modularity (integration) while high laterality of the whole brain corresponds to high modularity (segregation/functional specialization), our results also echo the recent findings that cognitive function depends on a dynamic, context-sensitive balance between functional integration and segregation [45–49].

In addition to LF and LR, we furthermore resolved the temporal structures of the whole-brain laterality dynamics. We revealed 3 recurring states (or "meta-states") with distinct laterality profiles, dwelling time, and transition probabilities. These states generally correspond to the multiple lateralization "axes" identified by Karolis and colleagues through dimensionality reduction of 590 meta-analysis maps: Our State 1 is characterized by strong left laterality of the left default mode and language networks, corresponding to the "symbolic communication" axis that involves Broca's and Wernicke's areas. Our State 3 demonstrates strong right laterality of frontoparietal and default mode networks in the right hemisphere consistent with the "active/perception" axis identified in [40].

Laterality of the human brain is hypothesized to be controlled by multiple factors [12], which is supported by the distinct spatial clusters identified by the clustering of laterality time series of all brain regions. Three of the 4 clusters (Clusters 1, 3, and 4) we identified roughly correspond to the top 3 factors identified in [12], which include the visual, attention, and the default mode network, respectively. It should be noted that our clustering analysis was conducted based on laterality dynamics of each individual, rather than on variations across individuals [12].

The 4 spatial clusters also showed distinct patterns of intercorrelations in their time-varying laterality index (Fig 3B). Of particular interest is that the time-varying laterality index of Cluster 4 (the default mode and language network) correlated negatively with those of the other 3 clusters (Cluster 1, visual network; Cluster 2, sensorimotor network; Cluster 3, attention and FPN network) in most participants (Clusters 4 and 1, 99.7% participants with significant negative correlation; Clusters 4 and 2, 99.1%; Clusters 4 and 3, 95.4%). This negative temporal correlation suggests opposite lateralization patterns between Cluster 4 and Clusters 1 to 3 over different time windows. The association analysis between dynamic lateralization and time-varying functional connectivity also suggested the essential role of Cluster 4: The increased level of right lateralization of Clusters 1/2/3 was associated with their decreased functional connectivity with Cluster 4 on the left hemisphere. Similarly, the increased level of left lateralization of Cluster 4 was associated with its decreased functional connectivity with Clusters 1/2/3 at the right hemisphere, suggesting that Cluster 4 might be critical for lateralization state of the whole brain. Furthermore, the more negative the correlation in laterality between Cluster 4 (default mode and language network) and the other 3 clusters (especially visual and sensorimotor network), the better the cognitive performance. This suggested that opposite lateralization pattern between Cluster 4 and Clusters 1 to 3 optimizes parallel processing in the 2 hemispheres [17] and enhances neural capacity. These results could be explained by the causal hypothesis of hemispheric specialization, which states that lateralization of one function forces the other function to the opposing hemisphere, which optimize parallel processing in complex tasks and increases processing efficiency [5,50,51].

Laterality dynamics of the brain may be related to multiple factors. Functionally, greater lateralization of higher-level brain regions was associated with disconnections with the contralateral hemisphere, lower BOLD activity of the whole brain, and greater network segregation, suggesting less exchange of information across hemispheres, which may be related to less energy (glucose) consumption [16,52]. Structurally, LF of a region correlated positively with its degree of structural connections. A brain region with wider structural connections may be modulated by multiple systems that likely demonstrate rich patterns of interhemisphere interaction, thus larger fluctuations in laterality [53,54]. Furthermore, we found that pairs of

regions with stronger structural connections are more positively correlated in their time-varying laterality index, indicating the role of structural connection in synchronization of functional lateralization of spatially distributed brain regions. In comparison, brain regions with negative intercorrelation of laterality showed less structural connections, possibly affected by more global factors, such as the regulatory effect of neurotransmitters on large-scale brain networks [39,47,55]. In view of the influence of GS and head movements on the dynamic brain network statistics [56,57], these 2 factors may also affect DLI. To exclude potential influence from these 2 factors, we repeated the above analyses by regressing GS and head movements parameters from the BOLD signal, respectively, and we found that our main results such as the distribution of DLI across the brain and their correlation with cognitive performance remain largely unchanged (see S11 and S12 Figs for details).

At the genetic level, structural brain lateralization is known to be heritable in large population samples [2]. Our research showed that the dynamic characteristics of lateralization were also heritable, especially in those high-order networks such as default mode and frontal-parietal networks, suggesting that laterality dynamics are stable properties of lateralization. From an evolutionary perspective, lateralization arises as a solution to minimize wiring costs while maximizing information processing efficiency in the rapid expansion of the cortex in evolution [58]. Higher-order systems such as default mode and frontal-parietal networks were among the most expanded regions in evolution [59]. Considering the dynamic laterality of these networks correlates significantly with cognitive performance, the heritability of dynamic lateralization in these networks, therefore, may confer evolutionary advantages.

## Limitations

In this study, resting-state data were analyzed, so it was not possible to directly relate the results to specific cognitive processes. Future studies should investigate dynamic changes in brain lateralization under task modulation, particularly with multiple task states or various task loads, to better understand the specific cognitive advantages of dynamic brain lateralization. In addition, while the GS-based laterality index has the advantages of being computationally convenient and highly correlated with the traditional ROI-based laterality index, it has the problem of averaging and being too coarse to detect network interactions. Therefore, we further analyzed which specific functional connectivity or network drives the temporal changes of laterality index of the 4 clusters. Finally, age and handedness are important factors potentially affecting lateralization, and handedness is also partly controlled by genetic factors [60]. Future studies are warranted that explore how cognitive deterioration with aging may affect lateralization dynamics.

## Conclusions

To conclude, our study demonstrates the dynamic nature of laterality in the human brain at resting state. We characterized comprehensively the spatiotemporal laterality dynamics by identifying 4 spatial clusters and 3 recurring temporal states and demonstrate that the temporal fluctuations of laterality as well as the negative correlation in laterality among different clusters associate with better language task and intellectual performance. We further explored the functional and structural basis underlying such laterality dynamics, and heritability of laterality dynamics. Our study not only contributes to the understanding of the adaptive nature of human brain laterality in healthy population but may also provide a new perspective for future studies of the genetics of brain laterality and potentially abnormal laterality dynamics in various brain diseases.

## Materials and methods

### Ethics statement

This paper utilized data collected for the HCP. The scanning protocol, participant recruitment procedures, and informed written consent forms, including consent to share deidentified data, were approved by the Washington University institutional review board [31]. Data acquisition for the HCP was approved by the Institutional Review Board of The Washington University in St. Louis (IRB # 201204036), and all open access data were deidentified. The data collected by the HCP adhered to the principles of the Declaration of Helsinki. No experimental activity with any involvement of human participants took place at the author's institutions. Our data analysis was performed in accordance with ethical guidelines of the Fudan University ethics committee.

### Dataset

We analyzed multimodal brain imaging data and behavioral measures from the HCP 1200 Subjects Release (S1200) [31]. All brain imaging data were acquired using a multiband sequence on a 3-Tesla Siemens Skyra scanner. For each participant, 4 resting-state fMRI scans were acquired: 2 with right-to-left phase encoding and 2 with left-to-right phase encoding direction (1,200 volumes for each scanning session, TR = 0.72 s, voxel size = $2 \times 2 \times 2$ mm). High-resolution T1-weighted MRI (voxel size = $0.7 \times 0.7 \times 0.7$ mm) and diffusion MRI (voxel size = 1.25 mm, 3 shells: $b$-values = 1,000, 2,000, and 3,000 s/mm$^2$, 90 diffusion directions per shell) were also acquired for each participant. A total of 991 participants (28.7 ± 3.7 years old, 528 females) were included in the final cohort utilized in this study according to the following criteria: (1) completed all 4 resting-state fMRI scans; (2) limited in-scanner head motion (mean FD < 0.2 mm); (3) same number of sampling points (i.e., 1,200 volumes per scanning session); (4) without any missing data in regions included in the employed parcellation scheme.

### Resting-state fMRI preprocessing

Resting-state fMRI data were preprocessed using the HCP minimal preprocessing pipeline (*fMRIVolume*) [61] and were denoised using the ICA-FIX method [62]. Then, a spatial smoothing (FHWM = 4 mm) and a low-pass filtering (0.01 Hz to 0.1 Hz) were performed. We did not employ GS regression, as the mean hemispheric time series were used for calculating the lateralization index [14]. The whole brain was parcellated into 360 regions (180 for each hemisphere), using the HCP's multimodal parcellation (HCP MMP1.0) [32]. These regions were grouped into 12 functional networks based on the CAB-NP [34].

### Dynamic laterality index

The pipeline for the DLI analysis is illustrated in Fig 1A. The DLI adopted a sliding window approach and a traditional laterality index [14] to capture the dynamics of laterality. DLI at the *t*th sliding time window is defined by:

$$DLI_t = r(ROI_i, GS_L) - r(ROI_i, GS_R),$$

where *ROIi* indicates BOLD time series of ROI *i*, and $GS_L$ and $GS_R$ indicate the GS, i.e., averaged time series of the voxels within the left and the right hemispheres, respectively. Pearson correlation coefficient *r* is Fisher-z-transformed. There are 3 main reasons why we adopted the hemispheric GS-based laterality: (1) GS has been shown to be able to reflect and characterize laterality by previous studies [14], and the computational complexity is very low (n*2, n being

the number of regions); (2) the whole-brain laterality pattern obtained using hemispheric GS is highly correlated to that obtained by ROI-based method like AI defined as follows [12,13,33,63]:

$$AI_i = \frac{\sum_1^n r(ROI_i, ROI_j)}{n} - \frac{\sum_1^m r(ROI_i, ROI_k)}{m}$$

where $\sum_1^n r(ROI_i, ROI_j)/n$ and $\sum_1^m r(ROI_i, ROI_k)/m$ indicate the averaged functional connectivity between BOLD time series of $ROI_i$ and all ROIs in the left and right hemisphere, respectively (S1 Fig), and n and m are number of regions in left and right hemisphere.

## Spatial clustering of laterality time series across the whole brain

To explore the spatial organization of dynamic lateralization across the whole brain, we performed spatial clustering on the laterality correlation matrix (obtained by calculating Pearson correlation between laterality time series of all pairs of brain regions). Specifically, we applied the Louvain community detection algorithm on the laterality correlation matrix averaged over all 991 participants and 4 runs. We set the γ parameter to 1, ran the algorithm for 100 times, and reported the cluster partitioning with the maximum modularity parameter ($Q$).

## Temporal clustering of whole-brain laterality state

To identify the recurring laterality states of the whole brain from 991 participants and 1,171 time windows, we adopted a 3-stage temporal clustering approach. Firstly, the $1,171 \times 4$ time windows of the 4 runs of each participant were clustered into 10 states by $k$-means clustering. Second, all 9,910 states (991 participants $\times$ 10 states) were clustered into 1,000 groups by a second level k-means clustering. Finally, we used Arenas-Fernandez-Gomez (AFG) community detection to cluster the 1,000 groups obtained in the second step. AFG allows for multiple resolution screening of the modular structure and a data-driving selection of clustering number. We determined the optimal number of clusters using the modularity coefficients obtained by changing the resolution parameter from 0.1 to 1.5 in steps of 0.1 [64]. The code of K-means and AFG clustering were from MATLAB function *kmeans* (with the cosine distance metric) and MATLAB Community Detection Toolbox (CDTB v. 0.9, https://www.mathworks.com/matlabcentral/fileexchange/45867-community-detection-toolbox) [65], respectively. Based on the final clustering, we took the average pattern of each category as centroids and reclassified all the windows of each participant according to the cosine distance between each window and each centroid.

## Cognitive performance in behavioral tasks

To understand the cognitive significance of the dynamic brain laterality, we utilized individual performance of a wide variety of cognitive tasks from the NIH Toolbox for Assessment of Neurological and Behavioral function and Penn computerized neurocognitive battery [66]. These involve story comprehension task that is more left lateralized, Flanker Task (inhibitory control and attention) that maybe more right lateralized, and Card Sort (cognitive flexibility) and List Sorting task (working memory) that tend to more bilateral.

The story comprehension task [67] includes a story condition that presents brief auditory stories adapted from Aesop's fables followed by a binomial forced-choice question to check the participants' understanding of the story topic (For example, after a story about an eagle that saves a man who had done him a favor, participants were asked, "Was that about revenge or reciprocity?"), and a math condition to answer addition or subtraction problems. To ensure similar level of difficulty across participants, math trials automatically adapted to the

participants' responses. The story and math trials were matched in terms of auditory, duration, attention demand, and phonological input. We used 3 relevant measures: "LanAcc" ("Language_Task_Story_Acc", the accuracy in answering questions about the story), "LanRT" ("Language_Task_Story_Median_RT", median response time to answer the questions), and "LanDiff" ("Language_Task_Story_Avg_Difficulty_Level", the average difficulty of all stories a participant can understand).

## Correlation between dynamic laterality index and cognitive performance

In calculating the correlation between dynamic laterality measures and cognitive performance (unadjusted scores), we used Permutation Analysis of Linear Models (PALM; http://fsl.fmrib. ox.ac.uk/fsl/fslwiki/PALM) to correct for the bias of significance estimation due to the kinship among participants. PALM used exchangeability blocks that is widely adopted to control the family structure–related bias in HCP data [68,69]. We calculated the Pearson correlation coefficients between dynamic lateralization attributes [MLI, LF, LR of 4 clusters; laterality correlation within and between 4 clusters; fraction and dwelling time of 3 states, state transitions] and cognitive performances, regressing out sex, age, education years, race, body mass index (BMI), handedness, gray matter volume, white matter volume, and head motion (mean FD). We obtained $p$-values for all correlation coefficients using 5,000 times permutation tests. FDR (q = 0.05) was used for multiple comparison correction (multiply comparison times = 4 MLI + 4 LF + 4 LR + 10 DLI correlation + 3 dwelling time +3 fraction + transition = 29). We show all the results of FDR q < 0.05. Moreover, we also highlight results that survive a more conservative multiple comparisons correction (FDR q < 0.05/13 = 0.0038, where 13 is the number or behavioral measurements used in this study).

## Association between dynamic laterality and dynamic functional connectivity

To identify which functional interactions may play important roles in the dynamic change of laterality, we adopted a regression-based approach to investigate which subnetworks (12 pairs of symmetric networks in CAB-NP) may affect the dynamic laterality of the 4 identified clusters. We averaged the functional connectivity within/between the 24 subnetworks; therefore, there are 24 * 23 / 2 = 276 subnetwork-level functional connectivity. Specifically, we built linear regression models of cluster-level DLI time series on subnetwork-level DFC and performed one-sample $t$ test on the regression coefficient (β), to investigate whether the influence of DFC on DLI was significantly greater than 0 (positive coupling) or less than 0 (negative coupling) (Fig 5A).

## Association between dynamic laterality and dynamic network properties and amplitude of low-frequency fluctuation

To understand the neural and network basis of dynamic brain lateralization, we correlated the laterality time series of each brain region with time-varying brain network measures (which reflect the organization of the brain like modular, or integration property) and the ALFF averaged over the whole brain for each participant. We constructed functional brain network using Pearson correlation within each sliding time window and computed the following topological measures:

1. Degree centrality ($DC$) [70,71], measuring the centrality of each node in the network:

$$DC_i = \sum_{j \in G} A_{ij}, i \neq j$$

where $G$ is a graph with the node $i$ and $j$ connected by edge $A_{ij}$.

2. Participation coefficient ($B_T$) measuring the contribution of each brain region to whole-brain integration:

$$B_{iT} = 1 - \sum_{s=1}\left(\frac{\kappa_{isT}}{\kappa_{iT}}\right)^2$$

where $\kappa_{isT}$ is the strength of the positive connections of region $i$ to regions belong to the module $s$ at time $T$; $\kappa_{iT}$ is the sum of strengths of all positive connections of region $i$ at time $T$. The participation coefficient of a region is close to 1 if its connections are distributed among all of the modules and 0 if all of its links are within its own module. Community division was obtained by community detection in each time window, and then the participation coefficient was calculated [25].

3. Modularity ($Q$) measuring the tendency of the network to segregate into several independent modules [72,73]:

$$Q = 1/(2m)\sum ij(Aij - kikj/(2m))\delta(Ci, Cj)$$

where $m$ is total number of edges in the network; $A$ is adjacent matrix; $k_i$ is degree of node $i$; $C_i$ is community assignment of node $i$; when and only when $C_i = C_j$, $\delta(C_i,C_j) = 1$, otherwise, $\delta(C_i,C_j) = 0$.

4. Interhemispheric connection [74], measuring the averaged interhemispheric connection strength.

Among these indicators, the averaged $DC$ is measured at the global network level, $B_T$ and $Q$ are measured at the module level, and interhemispheric connection is measured at the hemispheric level $DC$, $B_T$ and $Q$ were calculated using the Brain Connectivity Toolbox (BCT; http://www.brain-connectivity-toolbox.net/) [75]. We removed all negative connections in line with the general assumptions of graph theory [71].

ALFF measures spontaneous energetic activity within each time window [76] calculated as follows:

$$ALFF = 2 * \frac{abs(fft(BOLD))}{N}$$

where $fft$ is Fourier transform; $abs$ is operator of taking absolute value; $BOLD$ is BOLD signal within a time window in any region; and $N$ is the length of time window. At global level, we calculated the Pearson $r$ between the ALFF/network time series (averaged across the brain) and the absolute laterality time series (averaged across the brain). At local level, we correlated the ALFF/network time series (averaged across the brain) with the absolute laterality time series of each brain region.

## Structural network analysis

Because the probabilistic fiber tracking is very time-consuming, diffusion tensor imaging (DTI) data of a subset of HCP S1200 release, the "100 Unrelated Subjects" ($n = 100$, 54 females, mean age = 29 years) was used for the structural analysis (1 participant was removed due to head motion). Data underwent motion, susceptibility distortion, and eddy current distortion correction [61,77]. We used FMRIB Software Library v6.0 (FSL; https://fsl.fmrib.ox.ac.uk/fsl/fslwiki/) and *MRtrix3*, a toolkit for diffusion-weighted MRI analysis (https://mrtrix.readthedocs.io/en/dev/) [78], to construct the SC matrix. The processing steps were as follows: (1) tissue segmentation based on T1-weighted structural image; (2) calculation of 4D images

with gray matter, white matter, and cerebrospinal fluid using multi-shell, multi-tissue constrained spherical deconvolution [79]; (3) generation of white matter–constrained tractography using second-order integration over fiber orientation distributions (iFOD2), a probabilistic tracking algorithm (1,000,000 streamlines for each participant, max length = 250 mm, cutoff = 0.06) [80]; (4) use of FSL's FNIRT to reverse-register the MMP parcellation to individual space; (5) calculation of FA maps using *dtifit* of FSL; and (6) measurement of the average FA on all the streamlines connecting any 2 parcels. This process finally resulted in $360 \times 360$ FA matrices and FN matrices for 99 participants. We calculated the FA-degree and FN-degree based on this structural connectivity (SC) matrix, i.e., the sum of FA and FN of each row of the SC matrix. The Spearman correlation between laterality correlation matrix and SC matrix and the correlation between MLI/LF/LR and FN-/FA-degree were calculated for each participant.

## Heritability analysis

We used twin information in the HCP data to evaluate the heritability of the dynamic laterality measures. In the 991 participants, 103 pairs of MZ twins, 98 pairs of DZ twins, 195 pairs of SI were included, and their zygosity information was used to fit ACE model, which is able to divide total variance in a phenotype ($\sigma^2$) into 3 additive parts: (A) additive common genetic factor (heritability), (C) common environment, and (E) unshared environment, or other source of measurement error [81]:

$$\sigma^2 = A + C + E.$$

The ratio of the variance explained by A, C, and E to the total variance was used as estimates of heritability ($h^2$), environmental factors ($c^2$), and error ($e^2$), all of which sum to 1. All 4 models (for MLI, LF, LR, and laterality correlation matrix) were fitted and estimated using APACE (the Advanced Permutation inference for ACE models; www.warwick.ac.uk/tenichols/apace) [82]. Before model fitting, sex, age, education years, race, BMI, handedness, gray matter volume, white matter volume, and mean FD were regressed out from phenotype as covariables. The significance of heritability for each brain region was obtained using permutation test (1,000 permutations per region).

## Code availability

The code used in this study to calculate DLI is available on https://github.com/XinRanWu/Dynamic_Laterality.

## Supporting information

**S1 Fig. Repeatability of MLI and AI across different sessions.** Notice that the spatial distribution of MLI and AI is very similar. (A) Left: the average DLI map of 4 sessions of HCP data. Right: correlation coefficient among the 4 sessions (average among all the participants). (B) Left: the AI map calculated by 4 sessions of HCP data. Right: correlation coefficient between the 4 sessions (average among all the participants). The underlying data for this figure can be found in S5 Data. AI, autonomy index; DLI, dynamic laterality index; HCP, Human Connectome Project; MLI, mean laterality index.
(PNG)

**S2 Fig. Laterality correlation between homotopic regions (average of 991 participants).**
(PNG)

**S3 Fig.** (A) Dynamic laterality measures (MLI, LF, and LR) across 4 clusters. The statistical test was repeated-measure ANOVA, with the asterisk representing significance (***, $p < 0.01$, ****, $p < 0.001$) (B) The averaged laterality correlation matrix over all 991 participants (arranged by 12 functional networks of the CAB-NP). The underlying data for this figure can be found in S5 Data. CAB-NP, Cole-Anticevic Brain-wide Network Partition; LF, laterality fluctuations; LR, laterality reversal; MLI, mean laterality index.
(PNG)

**S4 Fig. The association between dynamic FC and the DLI time series of the 4 clusters.** We adopted the CAB-NP that contains 12 pairs of networks in bilateral hemispheres. We calculated the dynamic FC within/between the 12 pairs of networks, and then calculated their regression coefficients of DLI on FC for each participant in each run. We averaged the regression coefficients of the 4 runs of each participant and tested whether they were significantly greater than or less than 0 with single-sample $t$ test. The figure shows the $t$-values of each functional connectivity. Vis1, Visual1; vis2, Visual2; smn, Somatomotor; con, Cingulo-Opercular network; dan, Dorsal-Attention network; lan, Language network; fpn, Frontoparietal network; aud, Auditory network; dmn, Default Mode network; pmm, Posterior-Multimodal; vmm, Ventral-Multimodal; ora, Orbito-Affective. L, left hemisphere; R, right hemisphere. Note: This figure is the same as Fig 5 in the text, but it shows the regression coefficients of all FC (without thresholds) with more detail. The underlying data for this figure can be found in S4 Data. CAB-NP, Cole-Anticevic Brain-wide Network Partition; DLI, dynamic laterality index; FC, functional connectivity.
(PNG)

**S5 Fig.** (A-E) The association between the DLI and the average time series of network indicators and ALFFs of each brain region and each subnetwork. The values in the brain maps represent the $t$-value of the Pearson correlation coefficient r (one-sample $t$ test). The underlying data for this figure can be found in S5 Data. ALFF, amplitude of low-frequency fluctuation; DLI, dynamic laterality index.
(PNG)

**S6 Fig. The statistical differences of lateralization correlation matrix, ALFF, and network index between every pair of the 3 temporal states.** The paired $t$ test was used, and all values shown in the figure were t statistics. Only regions with $t$-value exceeded the significant threshold ($\pm2.33$, $p < 0.001$) were shown in brain maps. The underlying data for this figure can be found in S5 Data. ALFF, amplitude of low-frequency fluctuation.
(PNG)

**S7 Fig. The statistical differences of Q between every pair of the 3 temporal states.** The one-way ANOVA and post hoc test were used. One asterisk (*), $p < 0.05$; 2 asterisks (**), $p < 0.01$; 3 asterisks (***), $p < 0.001$; 4 asterisks (****), $p < 0.0001$; ns, nonsignificant. The underlying data for this figure can be found in S5 Data.
(PNG)

**S8 Fig. The association between the absolute value of averaged DLI time series of whole brain and the time-varying network indicators and ALFFs.** Higher absolute value represents higher left/right lateralization. We selected data from 3 participants, sub-100206, sub-100307, and sub-100408, to show the relationship between global lateralization (quantified by the averaged absolute value of laterality) and dynamic ALFF/graph theory indicators. The results show that higher global lateralization is associated with lower ALFF and stronger whole-brain dissociation (higher modularization Q, lower participant coefficient, centrality degree, and

interhemispheric connectivity). One asterisk (*), $p < 0.05$; 2 asterisks (**), $p < 0.01$; 3 asterisks (***), $p < 0.001$; 4 asterisks (****), $p < 0.0001$; ns, nonsignificant. The underlying data for this figure can be found in S5 Data. ALFF, amplitude of low-frequency fluctuation; DLI, dynamic laterality index.
(PNG)

**S9 Fig. The results of structural analysis.** (A) Mean structural connection matrix. (B) Mean FN degree. (C) FA degree. (D) The distribution of Spearman correlation coefficients between structural connection attributes [FN-degree, FA-degree, and SCM (based on FN or FA)] and dynamic laterality measures (LF, LR, and LCM). FA, fractional anisotropy; FN, fiber number; LCM, laterality correlation matrix; LF, laterality fluctuations; LR, laterality reversal; SCM, structural connection matrix.
(PNG)

**S10 Fig. The result of spilt-half validation.** (A-E) Results of part 1 (odd indexed participants in participant list, $N = 446$). (F-G) Results of part 2 (even indexed participants in participant list, $N = 445$). (A/F) MLI. (B/G) LF. (C/H) LR. (D/I) Result of spatial clustering. (E/J) Correlation between cognitive measures and indicator of dynamic lateralization. The cross indicates that P is less than 0.05 (FDR). C1-C4: correlation between DLI of Cluster 1 and Cluster 4, and so on. DLI, dynamic laterality index; FDR, false discovery rate; LF, laterality fluctuations; LR, laterality reversal; MLI, mean laterality index.
(PNG)

**S11 Fig. The results with GS of the whole brain being regressed out.** (A) The $t$-value of regression coefficient of DLI on GS. (B) MLI. (C) LF. (D) LR. (E) Result of spatial clustering. (F) The results of temporal clustering. (G) Correlation between cognitive measures and indicator of dynamic lateralization. The cross indicates FDR q < 0.05. C1-C4: correlation between DLI of Cluster 1 and Cluster 4, and so on. DLI, dynamic laterality index; FDR, false discovery rate; GS, global signal; LF, laterality fluctuations; LR, laterality reversal; MLI, mean laterality index.
(PNG)

**S12 Fig. The result using stricter head motion controlling procedure.** (A-F) Results using BOLD residual (with FD and Friston 24 parameters being regressed out). (A) MLI. (B) LF. (C) LR. (D) Result of spatial clustering. (E) Results of temporal clustering. (G-H) Correlation between cognitive measures and DLI. The cross indicates FDR q < 0.05. C1-C4: correlation between DLI of Cluster 1 and Cluster 4, and so on. (G) Correlation results using participants of mean FD < 0.15 mm. (H) Correlation results using participants of mean FD < 0.1 mm. DLI, dynamic laterality index; FD, frame distance; FDR, false discovery rate; LF, laterality fluctuations; LR, laterality reversal; MLI, mean laterality index.
(PNG)

**S13 Fig. The main result using different window size.** (A-E) Results of window size = 60 TRs. (F-G) Results of window size = 90 TRs. (A/F) MLI. (B/G) LF. (C/H) LR. (D/I) Result of spatial clustering. (E/J) Correlation between cognitive measures and DLI. The cross indicates that P is less than 0.05 (FDR corrected). C1-C4: correlation between DLI of Cluster 1 and Cluster 4, and so on. DLI, dynamic laterality index; FDR, false discovery rate; LF, laterality fluctuations; LR, laterality reversal; MLI, mean laterality index.
(PNG)

**S14 Fig. Repeatability of various dynamic laterality measures across different window lengths.** The r on vertical axis represents Pearson correlation coefficient between dynamic

laterality indictor patterns using window length of 60 TR/90 TR and indictor patterns of 30 TR (used in the text) among all participants. Each dot in the graph represents one of the 991 participants. As can be seen, MLI has the highest reproducibility (close to 1), followed by laterality correlation, LF, and LR. With the increase of the window length, the correlation between the dynamic laterality indicators and the results of the window length of 30 TR decreased gradually. REST1/REST2 indicates the scan time (in first or second day). LR/RL indicates the scanning direction. LR, left to right. RL, right to left. LF, laterality fluctuations; LR, laterality reversal; MLI, mean laterality index.
(PNG)

**S15 Fig.** (A-F) The influence of removing the ROI from its ipsilateral GS in calculating the laterality index. That is, we reconstructed GS by removing the ROI A from its ipsilateral GS. (A) MLI. (B) LF. (C) LR. (D) Result of spatial clustering. (E) The results of temporal clustering. (F) Correlation between cognitive measures and indicator of dynamic lateralization. The cross indicates FDR q < 0.05. C1-C4: correlation between DLI of Cluster 1 and Cluster 4, and so on. DLI, dynamic laterality index; FDR, false discovery rate; GS, global signal; LF, laterality fluctuations; LR, laterality reversal; MLI, mean laterality index; ROI, region of interest.
(PNG)

**S1 Table. The association between dynamic lateralization characteristic of 4 clusters and cognitive abilities.** Each row of the table corresponds to a cognitive test score in the HCP. PicVocab, Picture Vocabulary Test; ReadEng, Reading Test (reading decoding skill); Cardsort, Dimensional Change Card Sort Test (executive function, specifically tapping cognitive flexibility); Flanker, Flanker task (executive function, specifically tapping inhibitory control and attention); ProcSpeed, Pattern Comparison Processing Test (speed of processing); PicSeq, Picture Sequence Memory Test (episodic memory); VSPLOT_TC, Total Number Correct of Penn Line Orientation. PMAT24_A_CR, Number of Correct Responses of Penn Matrix Test (nonverbal reasoning); ListSort, List Sorting Working Memory Test; IWRD_TOT, Total Number of Correct Responses of Penn Word Memory; LanAcc, the accuracy of answering questions about the story of language task; LanRT, median response time to answer the questions of language task; LanDiff, the average difficulty of all stories of language task for each participant, which represents the overall language comprehension ability. r, Pearson correlation coefficient; p, the permutation test significance obtained by PALM. The bold text represents the association remains significant after FDR correction (q = 0.05, comparison number = 29). The asterisk (*) represents the association remains significant after FDR correction (q = 0.05/13 = 0.0038). See HCP Data Dictionary for more details. FDR, false discovery rate; HCP, Human Connectome Project; PALM, Permutation Analysis of Linear Models.
(XLSX)

**S2 Table. The association between laterality correlation among 4 clusters and cognitive abilities.** Each row of the table corresponds to a cognitive test score in the HCP. r, Pearson correlation coefficient; p, the permutation test significance obtained by PALM. The bold text represents the association remains significant after FDR correction (q = 0.05). The asterisk (*) represents the association remains significant after FDR correction (q = 0.05/13 = 0.0038). FDR, false discovery rate; HCP, Human Connectome Project; PALM, Permutation Analysis of Linear Models.
(XLSX)

**S3 Table. The association between characteristics of 3 laterality states and cognitive abilities.** Each row of the table corresponds to a cognitive test score in the HCP. r, Pearson correlation coefficient; p, the permutation test significance obtained by PALM. The bold text

represents the association remains significant after FDR correction (q = 0.05). The asterisk (*) represents the association remains significant after FDR correction (q = 0.05/13 = 0.0038). FDR, false discovery rate; HCP, Human Connectome Project; PALM, Permutation Analysis of Linear Models.
(XLSX)

**S1 Data. "S1_Data.xlsx" includes all individual observations used in Fig 2A.**
(XLSX)

**S2 Data. "S2_Data.xlsx" includes all individual observations used in Fig 2B.**
(XLSX)

**S3 Data. "S3_Data.xlsx" includes all individual observations used in Fig 2C.**
(XLSX)

**S4 Data. "S4_Data.xlsx" includes all statistic results (except for the statistics listed in S1– S3 Tables) and individual observations used in Figs 3–7.**
(XLSX)

**S5 Data. "S5_Data.xlsx" includes all statistic results and individual observations used in S1–S8 Figs.**
(XLSX)

**S1 Text. Supplemental text.**
(DOCX)

## Author Contributions

**Conceptualization:** Xinran Wu, Jie Zhang.

**Data curation:** Xinran Wu.

**Formal analysis:** Xinran Wu.

**Funding acquisition:** Jianfeng Feng, Jie Zhang.

**Methodology:** Xinran Wu, Xiangzhen Kong, Deniz Vatansever, Zhaowen Liu, Jie Zhang.

**Project administration:** Jianfeng Feng, Jie Zhang.

**Resources:** Jie Zhang.

**Supervision:** Xiangzhen Kong, Trevor W. Robbins, Jianfeng Feng, Jie Zhang.

**Validation:** Xinran Wu.

**Visualization:** Xinran Wu.

**Writing – original draft:** Xinran Wu.

**Writing – review & editing:** Xinran Wu, Xiangzhen Kong, Deniz Vatansever, Zhaowen Liu, Kai Zhang, Barbara J. Sahakian, Trevor W. Robbins, Paul Thompson, Jie Zhang.

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
