## [Editor Report · Decision Letter 0]

11 Nov 2021

Dear Jie, 

Thank you for submitting your revised manuscript entitled "Dynamic architecture of brain lateralization at rest" for consideration as a Research Article by PLOS Biology (previously PBIOLOGY-D-21-01985R1)

Your revised manuscript and response to reviewers have now been evaluated by the PLOS Biology editorial staff and the original academic editor, and I am writing to let you know that we would like to send your submission back to the reviewers. 

However, before we can do that, we need you to complete your submission by providing the metadata that is required for full assessment. To this end, please login to Editorial Manager where you will find the paper in the 'Submissions Needing Revisions' folder on your homepage. Please click 'Revise Submission' from the Action Links and complete all additional questions in the submission questionnaire.

Once your full submission is complete, your paper will undergo a series of checks in preparation for peer review. Once your manuscript has passed the checks it will be sent out for review. 

Please re-submit your manuscript within two working days, i.e. by Nov 15 2021 11:59PM.

Kind regards,

Luke

Lucas Smith

Associate Editor

PLOS Biology

lsmith@plos.org

---

## [Decision Letter · Decision Letter 1]

7 Dec 2021

Dear Dr Zhang,

Thank you for submitting a revised version of your manuscript "Dynamic architecture of brain lateralization at rest" for consideration as a Research Article at PLOS Biology. The revised version of your manuscript has been evaluated by the PLOS Biology editors, the Academic Editor and the original reviewers.

The reviews of your manuscript are appended below. As you will see both Reviewers 1 and 2 are satisfied by the revision. However, Reviewer 3 has raised a number of important concerns which will need to be thoroughly addressed before we can consider your manuscript for publication at PLOS Biology. Reviewer 3 has commented that the new analyses generated in the revision is presented suboptimally and that several of his/her previous requests were not adequately addressed. Having discussed Reviewer 3’s comments with the Academic Editor and with the other reviewers, we think that it would be essential for you to revise the manuscript to more clearly and carefully present the methods and results of your study, and to discuss more deeply how the study fits with or differs from previous work in the field. Moreover, we think that the analysis concerning lateralization in the context of task performance contained in the rebuttal should be included in the manuscript or supporting information. Additionally, we suggest the manuscript be carefully read and edited for English grammar to ensure that the manuscript is accessible to the widest possible audience.

In light of the reviews, we will not be able to accept the current version of the manuscript, but we would welcome re-submission of a much-revised version that addresses reviewer 3's comments. We cannot make any decision about publication until we have seen the revised manuscript and your response to the reviewer's comments. Your revised manuscript my be sent for further evaluation by the reviewers.

**IMPORTANT** In addition to carefully addressing the reviewer comments, please also address the following editorial requests: 

1 - ETHICS REQUEST: Please provide the protocol number which was approved by the Washington University IRB. If possible, please indicate whether the data collected by the Human Connectome Project adhered to the principles of the Declaration of Helsinki, or any other national or international guidelines

2 - DATA REQUEST: 

a- Supplementary files (e.g., excel). Please ensure that all data files are uploaded as 'Supporting Information' and are invariably referred to (in the manuscript, figure legends, and the Description field when uploading your files) using the following format verbatim: S1 Data, S2 Data, etc. Multiple panels of a single or even several figures can be included as multiple sheets in one excel file that is saved using exactly the following convention: S1_Data.xlsx (using an underscore).

b- Deposition in a publicly available repository. Please also provide the accession code or a reviewer link so that we may view your data before publication. 

Figure 2A-C; Fig 3A-D; Fig 4A-C; Fig 5A-B; Fig 6A-D; Fig S1A-C; Fig S3A-B; Fig S4A-E; Fig S5 A-C; Fig S6; Fig S7A-D; Fig S8

3 - DATA REQUEST: For studies involving third-party data, we encourage authors to share any data specific to their analyses that they can legally distribute. For any third-party data that the authors cannot legally distribute, they should include the following information in their Data Availability Statement upon submission:

A description of the data set and the third-party source

If applicable, verification of permission to use the data set

All necessary contact information others would need to apply to gain access to the data

Please ensure that the methods section of your manuscript contains the abovementioned informaton regarding the Human Connectome Project.

For more information on this policy, see here: https://journals.plos.org/plosbiology/s/data-availability

4 - METHODS SECTION - I noticed that some of your materials and methods section is included in the supplement. Please move all information on materials and methods into the manuscript.

5 - FIGURE LEGENDS: Please ensure that each figure legends contain enough information for the reader to understand the figure. In particular, please include information on statistical tests were used, and indicate within the figure any statistically significant changes (ex Supp Figure 8). When images contain stars to indicate significance (**), please define in the figure legend the threshold that each star represents. 

6 - TITLE: After some discussion within the team, we think that the title might be edited slightly to highlight the potential link to cognition. While we understand that the reviewers felt the original title should be toned down, if you agree, we think the title might be strengthened by changing it to something like the following: "Dynamic changes in brain lateralization correlate with human cognitive performance"

We expect to receive your revised manuscript within 1 month.

**IMPORTANT - SUBMITTING YOUR REVISION**

*Resubmission Checklist*

*Published Peer Review*

*Blot and Gel Data Policy*

Sincerely,

Lucas Smith

Associate Editor

PLOS Biology

lsmith@plos.org

REVIEWS:

Reviewer #1: I think the authors have done a nice job addressing previous reviewers comments. I have no further comments.

Reviewer #2: The authors have significantly improved upon the original submission. I am satisfied with the revised manuscript and the authors' thorough replies to my concerns.

Reviewer #3: The authors now report a considerable number of additional analyses that attempt to deal with the reviewers' comments. Unfortunately, the changes made to the manuscript were only very limited. The authors chose to bury the new analyses in the supplementary material, that now encompasses 13 Figures and the old tables. The text in the supplementary material is very preliminary, contains many orthographic mistakes and does not convey sufficient specific information on the added analyses. Also, the supplementary text and figures appear unstructured.

The most interesting aspect that arose from the revision was the new result that functional lateralization was more strongly related to diminished interhemispheric connectivity rather than increased intrahemispheric connectivity over time. 

I list only the most important points I see that were insufficiently addressed.

The control analyses are not sufficiently described. To name only a single point: How exactly did the authors regress the global signal out when one hemisphere's global signal was the critical time course for estimating functional connectivity with individual ROIs? Shouldn't this result in quasi zero correlation? Or did the authors regress out the average global signal over both hemispheres in their control analysis? I will let reviewer 2 comment on the movement issue; I am not entirely sure whether the authors convincingly dealt with her/his comment.

The authors still do not correct sufficiently for the number of tested correlations (neither the numbers of off-line behavioral measures seem to be considered, nor the number of the various correlation analyses in the different results subsections).

The authors did not change the speculative discussion on the relation between their findings in resting state and cognition. Mentioning the limitation of the study is not enough, when unjustified claims are made in the discussion. I appreciate the effort the authors made by analyzing task-related activity. Unfortunately, the authors did not find a relationship and argue in the rebuttal letter, why this could be the case. However, the only information I found on this issue in the revision, was the task description in the supplementary material. To improve the discussion, I suggest removing the speculations on cognition and rather focus on the claims the authors can make based on resting state data. 

Although the authors added some references to previous studies investigating lateralization of resting state dynamics, they did not put their findings into context. What do we learn from this new report compared to the previous ones? What did the old reports suggest that was disproven by the new data? Which aspects were replicated?

The authors did not explore well enough what their lateralization measure actually reflects. For example, the measure would not pick up a change in correlation between a left ROI and five right ROIs in case it is accompanied by a change in correlation between the same left ROI and five other right ROIs of the same magnitude. Because the choice of the lateralization measure is critical for the interpretation, I would at least have expected a clear statement what a correlation of a local signal with a global signal means, especially when the global signal of the ipsilateral hemisphere is "contaminated" more strongly by the ROIs activity that the other hemisphere's global signal. 

In sum, the expectation of a substantially revised manuscript that I got when reading the rebuttal letter was unfortunately not met.

---

## [Editor Report · Decision Letter 2]

13 Jan 2022

Dear Dr Zhang,

Thank you for submitting a revised version of your manuscript "Dynamic architecture of brain lateralization at rest" for consideration as a Research Article at PLOS Biology. I am handling your submission whilst my colleague Lucas Smith is out of the office. I am sorry for the delay in getting back to you on your revision due to the closure of the editorial office over the holiday period.

We have now discussed the revision with the academic editor, who is satisfied that the reviewer concerns have been fully addressed. Based on this, we will probably accept this manuscript for publication, provided you address the following data and other policy-related requests that I have provided below (points A-I).

(A) ETHICS REQUEST: Please provide the protocol number which was approved by the Washington University IRB. If possible, please indicate whether the data collected by the Human Connectome Project adhered to the principles of the Declaration of Helsinki, or any other national or international guidelines.

(B) DATA REQUESTS:

- Supplementary files (e.g., excel). Please ensure that all data files are uploaded as 'Supporting Information' and are invariably referred to (in the manuscript, figure legends, and the Description field when uploading your files) using the following format verbatim: S1 Data, S2 Data, etc. Multiple panels of a single or even several figures can be included as multiple sheets in one excel file that is saved using exactly the following convention: S1_Data.xlsx (using an underscore).

- Deposition in a publicly available repository. Please also provide the accession code or a reviewer link so that we may view your data before publication.

Figure 2A-C; Fig 3A-D; Fig 4A-C; Fig 5A-B; Fig 6A-D; Fig S1A-C; Fig S3A-B; Fig S4A-E; Fig S5 A-C; Fig S6; Fig S7A-D; Fig S8

(C) Please also ensure that figure legends in your manuscript include information on where the underlying data can be found, and ensure your supplemental data file/s has a legend.

(D) Please ensure that your Data Statement in the submission system accurately describes where your data can be found.

(E) For studies involving third-party data, we encourage authors to share any data specific to their analyses that they can legally distribute. For any third-party data that the authors cannot legally distribute, they should include the following information in their Data Availability Statement upon submission:

A description of the data set and the third-party source

If applicable, verification of permission to use the data set

All necessary contact information others would need to apply to gain access to the data

Please ensure that the methods section of your manuscript contains the abovementioned informaton regarding the Human Connectome Project.

For more information on this policy, see here: https://journals.plos.org/plosbiology/s/data-availability

(F) METHODS SECTION - I noticed that some of your materials and methods section is included in the supplement. Please move all information on materials and methods into the manuscript.

(G) FIGURE LEGENDS: Please ensure that each figure legends contain enough information for the reader to understand the figure. In particular, please include information on statistical tests were used, and indicate within the figure any statistically significant changes (ex Supp Figure 8). When images contain stars to indicate significance (**), please define in the figure legend the threshold that each star represents.

(H) TITLE: After some discussion within the team, we think that the title might be edited slightly to highlight the potential link to cognition. While we understand that the reviewers felt the original title should be toned down, if you agree, we think the title might be strengthened by changing it to something like the following: "Dynamic changes in brain lateralization correlate with human cognitive performance”

(I) The Academic Editor handling your submission has has also recommended the following textual edits to the following three sentences (edits recommended shown by addition of stars). 

“This study proposed a framework for dynamic laterality analysis. The time-averaged dynamic laterality index *replicated well* previous findings in terms of the basic (static) spatial patterns of functional lateralization, while the time varying laterality measures provided new insights into the dynamic changes of lateralization in resting-state activity.”

This suggested laterality in lower-order regions is more flexible, consistent with previous research on the intrinsic time scales of higher and lower-order regions. That is, the bilateral sensory areas need to process constantly changing sensory inputs and integrate them with information from the contralateral hemisphere in real time to form accurate and coherent *percepts*”

Figure 5. Association between dynamic laterality time series of four clusters and dynamic functional connectivity within/between 24 subnetworks. (a) Analysis pipeline. We established linear regression models of cluster-level DLI time series on sub-network level dynamic functional connectivity (DFC) and performed one-sample t-tests on the regression coefficients (β) to investigate whether the influence of DFC on DLI was significantly greater than 0 (positive coupling) or less than 0 (negative coupling). (b) the association between the dynamic laterality time series of Cluster 1/2/3/4 and the DFC within/between 24 subnetworks (according to Cole-Anticevic Brain-wide Network Partition). Only functional connections with highly significant t values for regression (with p<1e-200 or |t|>40, one-sample t-test) are shown since otherwise there are too many significant connections. Here *a* positive laterality index indicates left-lateralization. Colors of points indicate the network arrangement of the Cole-Anticevic Brain-wide Network Partition (CAB-NP). Vis1, Visual1; vis2, Visual2; smn, Somatomotor; con, Cingulo-Opercular network; dan, Dorsal-Attention network; lan, Language network; fpn, Frontoparietal network; aud, Auditory network; dmn, Default Mode network; pmm, Posterior-Multimodal; vmm, Ventral-Multimodal; ora, Orbito Affective. L, left hemisphere; R, right hemisphere

We expect to receive your revised manuscript within two weeks. 

*Published Peer Review History*

*Early Version*

Sincerely,

Richard 

Richard Hodge, PhD

Associate Editor, PLOS Biology

rhodge@plos.org

On behalf of:

Lucas Smith, Ph.D

Associate Editor, PLOS Biology

lsmith@plos.org

---

## [Editor Report · Decision Letter 3]

31 Jan 2022

Dear Dr Zhang,

On behalf of my colleagues and the Academic Editor, Matthew Rushworth, I am pleased to say that we can in principle accept your Research Article "Dynamic changes in brain lateralization correlate with human cognitive performance" for publication in PLOS Biology, provided you address any remaining formatting and reporting issues. These will be detailed in an email that will follow this letter and that you will usually receive within 2-3 business days, during which time no action is required from you. Please note that we will not be able to formally accept your manuscript and schedule it for publication until you have any requested changes.

PRESS

Sincerely,

Richard

Richard Hodge, PhD

Associate Editor, PLOS Biology

rhodge@plos.org

On behalf of:

Lucas Smith, PhD

Associate Editor, PLOS Biology

lsmith@plos.org